# Dual-atom Pt heterogeneous catalyst with excellent catalytic performances for the selective hydrogenation and epoxidation

Shubo Tian[1,2], Bingxue Wang[3], Wanbing Gong[4], Zizhan He[3], Qi Xu[2], Wenxing Chen[5], Qinghua Zhang[6], Youqi Zhu[5], Jiarui Yang[2], Qiang Fu [3✉], Chun Chen[4], Yuxiang Bu [3], Lin Gu[6], Xiaoming Sun[1], Huijun Zhao[4], Dingsheng Wang [2✉] & Yadong Li[2]

Atomically monodispersed heterogeneous catalysts with uniform active sites and high atom utilization efficiency are ideal heterogeneous catalytic materials. Designing such type of catalysts, however, remains a formidable challenge. Herein, using a wet-chemical method, we successfully achieved a mesoporous graphitic carbon nitride (mpg-$C_3N_4$) supported dual-atom $Pt_2$ catalyst, which exhibited excellent catalytic performance for the highly selective hydrogenation of nitrobenzene to aniline. The conversion of ˃99% is significantly superior to the corresponding values of mpg-$C_3N_4$-supported single Pt atoms and ultra-small Pt nanoparticles (~2 nm). First-principles calculations revealed that the excellent and unique catalytic performance of the $Pt_2$ species originates from the facile $H_2$ dissociation induced by the diatomic characteristics of Pt and the easy desorption of the aniline product. The produced $Pt_2$/mpg-$C_3N_4$ samples are versatile and can be applied in catalyzing other important reactions, such as the selective hydrogenation of benzaldehyde and the epoxidation of styrene.

[1] State Key Laboratory of Chemical Resource Engineering, Beijing Advanced Innovation Centre for Soft Matter Science and Engineering, Beijing University of Chemical Technology, Beijing, China. [2] Department of Chemistry, Tsinghua University, Beijing, China. [3] School of Chemistry and Chemical Engineering, Shandong University, Jinan, China. [4] Key Laboratory of Materials Physics, Centre for Environmental and Energy Nanomaterials, Anhui Key Laboratory of Nanomaterials and Nanotechnology, Institute of Solid State Physics, Chinese Academy of Sciences, Hefei, China. [5] Beijing Key Laboratory of Construction Tailorable Advanced Functional Materials and Green Applications, School of Materials Science and Engineering, Beijing Institute of Technology, Beijing, China. [6] Institute of Physics, Chinese Academy of Sciences, Beijing, China. ✉email: qfu@sdu.edu.cn; wangdingsheng@mail.tsinghua.edu.cn

Atomically monodispersed heterogeneous catalysts with uniform active sites can be used as ideal models for understanding the correlations between compositions/structures and the corresponding performances, which are not only significant but also challenging in heterogeneous catalysis research[1–4]. Besides, atomically monodispersed catalysts also have high atom utilization efficiency and usually exhibit excellent catalytic activity[5–9]. For example, the well-known single-atom catalysts have been widely employed in many heterogeneous reactions[10–25]. Compared with the single-atom catalysts, dual-atom catalysts not only possess the same advantages of uniformity in the active sites and high atom utilization efficiency[26–29], the involved two metal atoms can also cooperate and play a synergistic role in optimizing interactions between the active sites and the reactants or intermediates[30–35]. This may help to break the intrinsic linear scaling relationships between adsorption energies of reaction intermediates and further improve the catalytic performances. Although dual-atom heterogeneous catalysts possess so many unique advantages, synthesizing such materials remains a great challenge, which mainly comes from the difficulty in controlling the configuration uniformity of the active sites at the atomic scale.

Selective hydrogenation and epoxidation are two significant approaches to produce fine chemicals and high-value products in practical industrial applications[36,37]. These reactions include, for instance, the selective hydrogenation of nitroarenes to amines[38,39], the selective hydrogenation of aldehyde compounds to alcohol compounds[40,41], and the epoxidation of alkenes to epoxides[42,43]. So far, many non-noble metal catalysts have been developed for catalyzing the reactions, but the harsh reaction conditions hinder their wide applications[44,45]. Therefore, conventional noble metal catalysts are still the most commonly employed catalysts in those reactions, but they generally suffer from the low efficiency of atom utilization and the inevitably high cost[43–46]. We thereby expect that atomically monodispersed heterogeneous catalysts can play a role in the reactions.

Herein, we have successfully synthesized a Pt$_2$/mpg-C$_3$N$_4$ catalyst by using a simple wet-chemical method. The as-prepared sample possessed a dual-atom Pt$_2$ structure that was evidenced with aberration correction transmission electron microscopy (TEM), X-ray absorption fine structure data, and first-principles simulations. The dual-atom Pt species exhibited excellent catalytic performance toward the selective hydrogenation of nitrobenzene to aniline, and behaved much better than the corresponding Pt single-atom catalysts and Pt nanoparticles (~2 nm). First-principles calculations revealed that the unique catalytic properties of Pt$_2$/mpg-C$_3$N$_4$ originate from the easily breaking of the H–H bond in the H$_2$ reactant and the effective release of the aniline product. The application of the superior Pt$_2$/mpg-C$_3$N$_4$ catalyst has also been extended to the selective hydrogenation of benzaldehyde to benzyl alcohol and the epoxidation of styrene to styrene oxide, demonstrating the versatility of the dual-atom Pt species in heterogeneous catalysis.

## Results

**Synthesis and characterization of Pt$_2$/mpg-C$_3$N$_4$ samples.** The Pt$_2$/mpg-C$_3$N$_4$ sample was synthesized using the wet-chemical strategy. (Ethylenediamine)iodoplatinum(II) dimer dinitrate and mesoporous graphitic carbon nitride (mpg-C$_3$N$_4$) were selected as the dual-atomic Pt precursor and the substrate. They were mixed and further pyrolyzed to remove the ligands from the dual-atom Pt precursor. The Pt$_1$/mpg-C$_3$N$_4$ and the Pt nanoparticle/mpg-C$_3$N$_4$ samples were synthesized using the same method, except that the Pt species had been replaced by H$_2$PtCl$_6$ and Pt nanoparticles, respectively (see "Methods" for more details).

The X-ray diffraction (XRD) pattern (Supplementary Fig. 1) demonstrated that the synthesized mpg-C$_3$N$_4$ sample has a graphitic packing structure[47,48], and the disordered spherical pores of mpg-C$_3$N$_4$ were captured by the TEM image (Supplementary Fig. 2). Upon the loading of the dual-atom Pt precursor, neither Pt nanoparticles nor nanoclusters were observed in the TEM (Supplementary Fig. 3) and high-angle annular dark field scanning transmission electron microscopy (HAADF-STEM) images (Fig. 1a). Moreover, no additional diffraction peak of the Pt lattices was found in the XRD pattern (Supplementary Fig. 1). The energy-dispersive X-ray (EDX) spectroscopy further demonstrated a homogeneous distribution of the Pt species (Fig. 1b). The results from the inductively coupled plasma optical emission spectrometry estimated that the content of Pt is ~0.15 wt%. All the above results indicated that dual-atom Pt had been homogeneously dispersed on the mpg-C$_3$N$_4$ substrate. After the pyrolysis procedure, the infrared absorption peaks that correspond to the ligands of the precursor (at ~580, 825, 1005, 1050, 1340, 3090, and 3230 cm$^{-1}$) were not observed in the Pt$_2$/mpg-C$_3$N$_4$ sample, which supported a complete removal of the ligand molecules (Supplementary Fig. 4). To further confirm the dual-atomic feature of the Pt species, aberration-corrected (AC) HAADF-STEM was applied to characterize the Pt$_2$/mpg-C$_3$N$_4$ sample. Many paired bright dots (marked with white circles) were observed in the AC HAADF-STEM image, which is consistent with the feature of two Pt atoms (Fig. 1c). Here, depending on the orientation of the Pt–Pt bond relative to that of the incident beam direction, the appearance of the paired bright dots can be different from each other (Supplementary Fig. 5), because the AC HAADF-STEM imaging only represents a two-dimensional projection of the three-dimensional Pt$_2$/mpg-C$_3$N$_4$ samples[49]. Besides, a few isolated bright dots (marked with green circles) were also observed, which we attributed to an overlap of the two Pt atoms in the incident beam direction, or, to the incomplete imaging of the dual-atom Pt species owing to an incomplete focusing (Fig. 1c). To confirm the

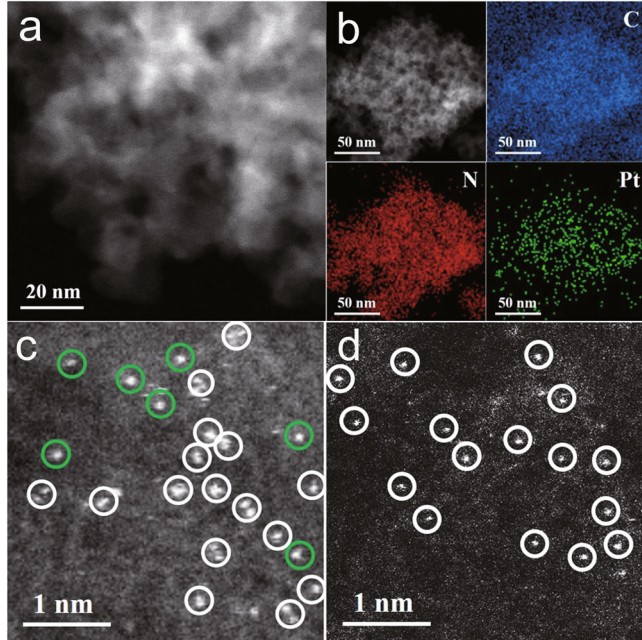

**Fig. 1 Characterization of Pt$_2$/mpg-C$_3$N$_4$ sample. a** HAADF-STEM image of Pt$_2$/mpg-C$_3$N$_4$. **b** EDX mapping distributions of the C (blue), N (red), and Pt (green) elements, respectively. **c, d** AC HAADF-STEM images of the Pt$_2$/mpg-C$_3$N$_4$ and Pt$_1$/mpg-C$_3$N$_4$ samples, respectively.

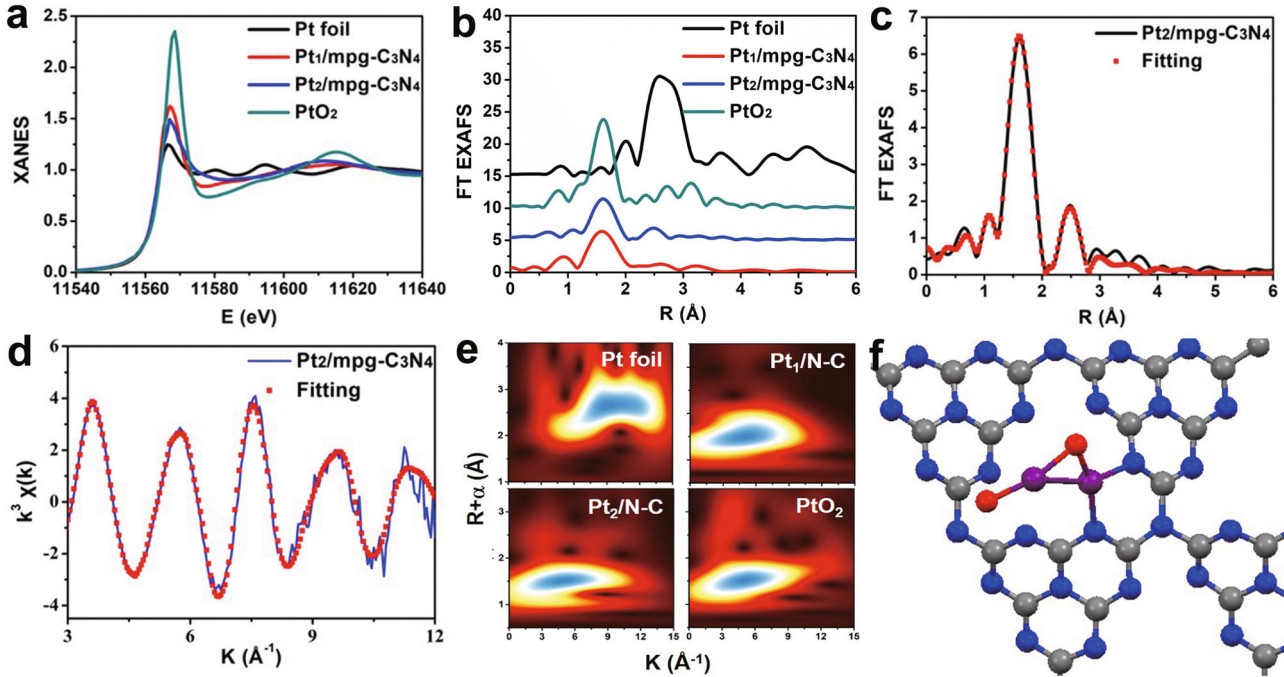

**Fig. 2 Pt $L_3$-edge XAFS analysis. a**, **b** XANES and FT EXAFS spectra of Pt$_2$/mpg-C$_3$N$_4$, Pt$_1$/mpg-C$_3$N$_4$, and corresponding references. **c**, **d** The FT EXAFS fitting spectrum of Pt$_2$/mpg-C$_3$N$_4$ at $R$- and $k$-space, respectively. **e** WT EXAFS of Pt foil, Pt$_1$/mpg-C$_3$N$_4$, Pt$_2$/mpg-C$_3$N$_4$, and PtO$_2$. **f** The schematic model of Pt$_2$/mpg-C$_3$N$_4$ (C: gray; N: blue; O: red; Pt: purple).

existence of the latter factor, we compared the different images that were collected within the same region of the sample, but with the focus of the imaging constantly changed (Supplementary Fig. 6). It can be seen that under different focusing conditions, many isolated dots can indeed be imaged as paired dots, demonstrating that the images actually come from the dual-atom Pt species. By contrast, the Pt$_1$/mpg-C$_3$N$_4$ sample always exhibits and maintains the characteristics of a single bright spot (except under special circumstances as presented in Supplementary Fig. 11), further confirming the sharp difference between the dual-atom and the single-atom Pt species (Fig. 1d and Supplementary Figs. 7–10).

X-ray absorption fine structure spectroscopy, which is a powerful technique for determining the chemical state and coordinated environment, was also applied to characterize the Pt species. The Pt $L_3$-edge X-ray absorption near-edge structure (XANES) spectra of the Pt$_2$/mpg-C$_3$N$_4$ and Pt$_1$/mpg-C$_3$N$_4$ samples, as well as the corresponding references, are shown in Fig. 2a. Here, the white line intensity peak of dual-atom Pt$_2$ is located between those of Pt foil and PtO$_2$, indicating that the two Pt atoms possess positive charges. This can be attributed to the strong interaction between the dual-atom Pt species and the mpg-C$_3$N$_4$ substrate or partial oxidation of dual-atom Pt by the O$_2$, which is similar to the case of Pt$_1$/mpg-C$_3$N$_4$. The Fourier-transformed (FT) $k^3$-weighted extended X-ray absorption fine structure (EXAFS) spectra of Pt$_2$/mpg-C$_3$N$_4$ showed a sharp peak located at 1.62 Å, which is also similar to the result of Pt$_1$/mpg-C$_3$N$_4$ and can be assigned to the Pt–N/O contributions (Fig. 2b). For Pt$_2$/mpg-C$_3$N$_4$, another distinct peak at 2.44 Å was found, similar to the Pt–Pt path of Pt foil, but not observed in the spectrum of the Pt$_1$/mpg-C$_3$N$_4$ sample. It reveals that the Pt–Pt path should also be taken into account in the spectrum of Pt$_2$/mpg-C$_3$N$_4$. The wavelet transforms (WT) EXAFS analysis, which can discern scattering atoms and provide both $R$-space and $k$-space resolutions, was also employed (Fig. 2e). Here, the WT EXAFS spectrum of the Pt$_2$/mpg-C$_3$N$_4$ sample showed a

maximum at 4.6 Å$^{-1}$ in $k$-space and at 1.6 Å in $R$-space, which we attributed to the Pt–O/N bonds. Besides, there were two distinct peaks at ~2.5 Å in $R$-space ($k = 8$ and 3.7 Å$^{-1}$, respectively), which came from the contribution of the Pt–Pt bond. Especially, the peak at $k = 3.7$ Å$^{-1}$ means that there were oxygen atoms connecting with Pt. According to these results, the structure of Pt$_2$ consists of a Pt–Pt bond with surrounding O attached. The EXAFS fitting results further showed that the first peak at 1.62 Å comes from the Pt–N/O contributions and the second one at 2.44 Å is from the Pt–Pt path (Fig. 2c, d, Supplementary Figs. 12 and 13, and Supplementary Table 1). The coordination number of the Pt–N/O path was estimated to be 2.4 at the distances of 2.02 Å, and the coordination number from the second sphere by the Pt–Pt path was assessed to 1.1, with the corresponding distance being 2.61 Å.

The configuration of the Pt$_2$/mpg-C$_3$N$_4$ sample was also explored by extensive first-principles calculations, with the optimized geometry shown Fig. 2f. As shown in Supplementary Fig. 14, the graphitic carbon nitride (g-C$_3$N$_4$) substrate is distorted showing obvious undulations. We first considered various kinds of Pt$_2$/g-C$_3$N$_4$ structures without oxygen atoms (Supplementary Fig. 15) in the simulations, but none of them matched the XAFS information. It is worth noting that, since the measurements of the XAFS spectra were performed in air, the oxygen molecules contained in the atmosphere had interacted with and attached to the Pt species, which is also consistent with the WT EXAFS analysis result (Fig. 2e). We note in passing that such oxygen atoms will be removed by hydrogen molecules at the initial stage of the hydrogenation reactions discussed in the next section. We then explored the possible structures of dual-atom Pt$_2$ with oxygen attached (Supplementary Fig. 16) and found one configuration (Fig. 2f) that agrees well with the results from the XAFS data. Here, the Pt–N/O and the Pt–Pt bond lengths were calculated to be 1.96 ± 0.14 and 2.55 Å, with the corresponding coordination numbers being 2.5 and 1.0, respectively, based on the model in Fig. 2f.

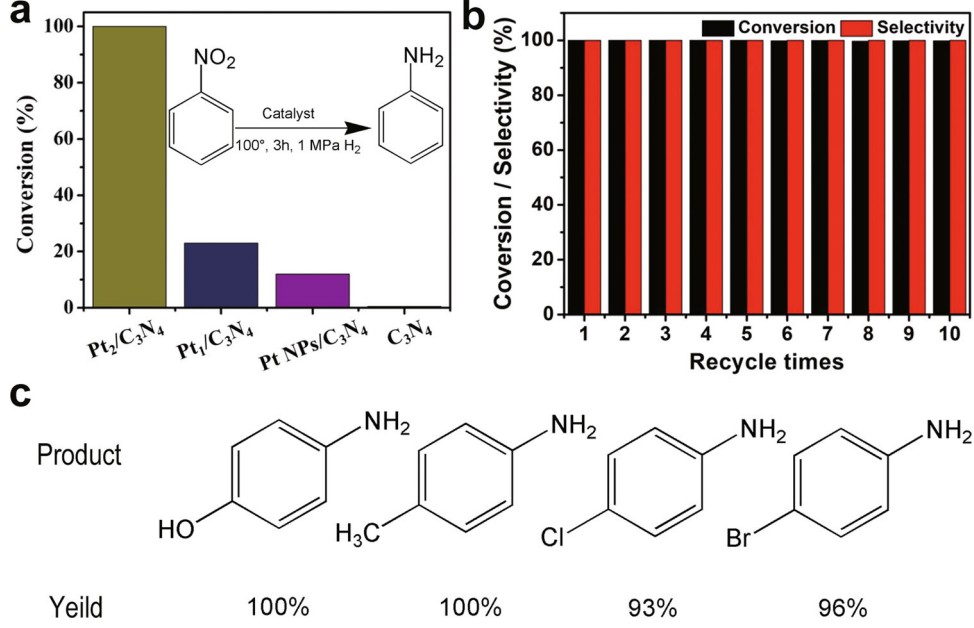

**Fig. 3 Hydrogenation of nitrobenzene. a** Catalytic performance for the hydrogenation of nitrobenzene by $Pt_2/mpg-C_3N_4$ and other reference samples. **b** Recycling of $Pt_2/mpg-C_3N_4$ for the catalytic hydrogenation of nitrobenzene. **c** Hydrogenation of functionalized nitroarenes catalyzed by $Pt_2/mpg-C_3N_4$.

**Hydrogenation of nitrobenzene to aniline.** The catalytic performance of $Pt_2/mpg-C_3N_4$, $Pt_1/mpg-C_3N_4$, and Pt nanoparticles (~2 nm)/mpg-$C_3N_4$ (Supplementary Fig. 17) were then investigated. Under the conditions of 1 MPa $H_2$ and 3 MPa $N_2$ pressure at 100 °C, a conversion of >99% was obtained on the $Pt_2/mpg-C_3N_4$ catalyst for the hydrogenation of nitrobenzene to aniline, while no by-product was detected (Fig. 3a). Such $Pt_2/mpg-C_3N_4$ sample can be reused at least ten times without any loss of the activity (Fig. 3b). After ten cycles, the AC HAADF-STEM image and the EXAFS spectrum of the $Pt_2/mpg-C_3N_4$ material did not exhibit any change, indicating that the Pt species were still well dispersed as dual-atom Pt pairs (Supplementary Figs. 18 and 19). It may be worth mentioning that the mpg-$C_3N_4$ support is reactively inert under the same conditions. The corresponding conversions of $Pt_1/mpg-C_3N_4$ and Pt nanoparticles/mpg-$C_3N_4$, by contrast, sharply dropped to 23% and 12%, respectively, demonstrating the uniqueness of the dual-atom Pt species in the catalytic properties (Fig. 3a). After the reaction, the Pt species in $Pt_1/mpg-C_3N_4$ was still well dispersed as single atoms (Supplementary Fig. 20), and the size of the Pt nanoparticles on mpg-$C_3N_4$ did not change either (Supplementary Fig. 21), indicating that both $Pt_1$ and Pt nanoparticles were stable in the catalytic process. To investigate whether the outstanding catalytic performance of the dual-atom $Pt_2$ catalyst is general in the hydrogenation of nitroarenes, we have also explored the hydrogenation of several other nitroarene derivatives, including p-nitrophenol, p-nitrotoluene, tetrachloro-nitrobenzene, and tetrabromonitrobenzene. We found that $Pt_2/mpg-C_3N_4$ exhibited excellent yields toward all corresponding anilines (Fig. 3c and Supplementary Table 2).

The changes in the oxidation state and chemical bonding of the $Pt_2$ species during the process of the nitrobenzene hydrogenation were also detected by the time-dependent XAFS (Supplementary Figs. 22 and 23). The Pt $L_3$-edge in the XANES spectra of $Pt_2/mpg-C_3N_4$ showed that the intensity of the white line peaks became lower during the reaction (Supplementary Fig. 22), meaning that the oxidation state of Pt was smaller than that in the initial state. It is not surprising because the oxygen atoms attached to Pt can be removed by the hydrogen molecules. Besides, the EXAFS spectra showed that the first shell peak shifted

from 1.57 to 1.55 Å (Supplementary Fig. 23), indicating that shorter chemical bonds like that of Pt–H appeared in the reaction process. Herein, it may be worth mentioning that the EXAFS exhibits a high sensitivity to the bond length change between center metal atoms and neighboring atoms, even when the change is as small as 0.01 Å (ref. [50]). Although the shift of the first shell peak during nitrobenzene hydrogenation is slight, it can indeed be monitored by EXAFS.

To understand the underlying reason for the unique and excellent catalytic properties of the dual-atom $Pt_2$ system, we have performed systematic first-principles simulations. It is found that the excellent and unique catalytic performance of the $Pt_2$ species originates from the facile dissociation of the $H_2$ reactant, which is induced by the diatomic characteristics of Pt, and the easy desorption of the aniline product. In Fig. 4 and Supplementary Fig. 24, we present the reaction pathway and the computational energy profile of the entire nitrobenzene hydrogenation process on the $Pt_2/g-C_3N_4$ catalyst, which starts from the adsorption of the nitrobenzene reactant, and ends with the desorption of the aniline product and the regeneration of the $Pt_2$ active site. As can be seen in Fig. 4, the nitrobenzene molecule connects with the two Pt atoms via the two oxygen atoms of the nitro group, showing an adsorption energy of −3.16 eV (a negative value means exothermic adsorption). Subsequently, one of the N–O bonds breaks (S1 → S2 via TS1; TS is short for transition state), and the corresponding energy barrier is only 0.58 eV. It is not surprising that the energy barrier is so low, since the N–O bonds have been well activated upon the nitrobenzene adsorption. In the Supplementary Fig. 26, we present the calculated electronic density of state (DOS) of $Pt_2/g-C_3N_4$ upon the adsorption of nitrobenzene. One can see that the Pt diatomic species and the nitrobenzene adsorbate interact with each other at the Fermi level. By analyzing the corresponding spatial distribution, we find that the electronic state comes from the lowest unoccupied molecular orbital (LUMO) of nitrobenzene. Since the LUMO of nitrobenzene involves the antibonding interactions of the π orbitals between the N and the two O atoms (inset in the Supplementary Fig. 26), the facts that this LUMO appears at the Fermi level and is partially occupied upon nitrobenzene adsorption will inevitably bring about the weakening of the N–O

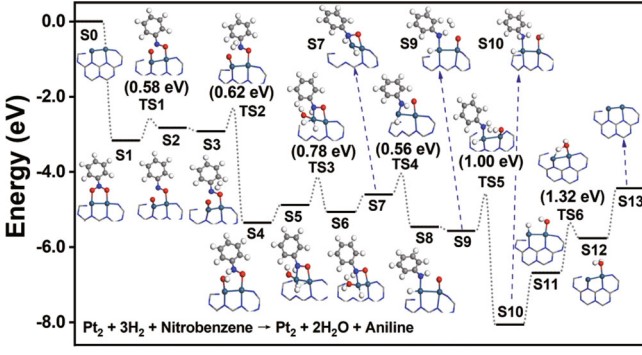

**Fig. 4 First-principles calculations of hydrogenation of nitrobenzene on Pt$_2$/g-C$_3$N$_4$.** Reaction pathway and computational energy profile of nitrobenzene hydrogenation on the Pt$_2$/g-C$_3$N$_4$ catalyst. The label S0 represents the initial state and the subsequent labels S1–S12 represent a series of intermediate states. The labels TS1–TS6 (TS is short for transition states) represent a series of transition states. Here, only the key structures, i.e., the Pt$_2$ catalytic system, as well as the adsorbate bound on it, are shown. The information regarding reactant molecules which have not been adsorbed, and/or product molecules which have been desorbed are labeled in the Supplementary Information (Supplementary Fig. 24). The teal, gray, blue, red, and white spheres represent the Pt, C, N, O, and H atoms, respectively.

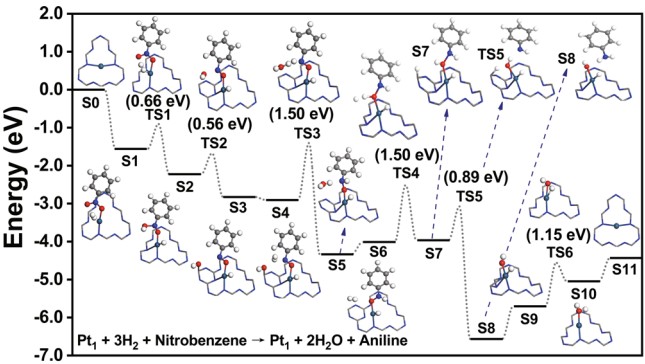

**Fig. 5 First-principles calculations of hydrogenation of nitrobenzene on Pt$_1$/g-C$_3$N$_4$.** Reaction pathway and computational energy profile of nitrobenzene hydrogenation on the Pt$_1$/g-C$_3$N$_4$ system. The label S0 represents the initial state and the subsequent labels S1–S10 represent a series of intermediate states. The labels TS1–TS6 (TS is short for transition states) represent a series of transition states. Here, only the key structures, i.e., the Pt$_1$ catalytic system as well as the adsorbate bound on it, are shown. The information regarding reactant molecules which have not been adsorbed, and/or product molecules which have been desorbed are labeled in the Supplementary Information (Supplementary Fig. 27). The teal, gray, blue, red, and white spheres represent the Pt, C, N, O, and H atoms, respectively.

bonds and the elongation of the corresponding bond lengths from 1.25 Å (the corresponding value of an isolated nitrobenzene molecule) to 1.35 Å, both of which lead to the easily breaking of the N–O bonds, and the generation of the unsaturated N and O adsorbates.

Interestingly, the unsaturated N and O adsorbates on each of the two Pt atoms can effectively promote the dissociation of the H$_2$ reactant. In Fig. 4, one can see that after overcoming an energy barrier of only 0.62 eV (S3 → S4 via TS2), the first H$_2$ molecule reacts with the N and the O atoms via the Eley–Rideal (ER)

mechanism, producing the first N–H bond and a hydroxyl group. Here, the activation of the H$_2$ molecule does not involve the participation of Pt atoms, reminiscent of the nitrogen-doped carbon nanotube arrays as metal-free electrocatalysts for the oxygen reduction reaction[51]. In the Supplementary Fig. 25, we present the calculated electronic DOS of the two H atoms that are involved in the H$_2$ dissociation for the configurations S3 and TS2 in Fig. 4. One can see that while in S3 (left panel in the Supplementary Fig. 25), the two H atoms have the same electronic structure that is very close to that of the hydrogen atoms in an isolated H$_2$ molecule, in TS2 (right panel in the Supplementary Fig. 25), the electronic structures of the two H atoms are significantly different. In particular, at the peaks of 0.39 and 0.09 eV below the Fermi level, the H atom close to the N atom (with the H–N distance being 1.87 Å) exhibits an obvious electronic state distribution, while the H atom close to O (with the H–O distance being 1.52 Å) has almost no distribution. Such remarkable contrast can also be visualized from the spatial distributions of the electronic states within the corresponding energy intervals (inset in the right panel). Here, one of the two H atoms as well as the unsaturated N and O adsorbates exhibits the distributions. The results indicate that the H$_2$ activation is facilitated by the produced N and O atoms, and is promoted by a polarization effect induced by O. The latter is further supported by the Bader charge analysis, showing that in TS2, the two H atoms carry charges of +0.20 (close to O) and −0.08 (close to N), respectively.

From S5 to S6, the second H$_2$ molecule adsorbs on one Pt atom and then dissociates after overcoming a barrier of 0.78 eV (TS3). Upon desorption of the produced water molecule (S6 to S7), the second N–O bond breaks in a similar way, showing an energy barrier of 0.56 eV (S7 → S8 via TS4). As the third H$_2$ molecule participates in the reaction via the same ER mechanism, the second N–H bond forms and the aniline molecule is produced after overcoming an energy barrier of 1.00 eV (S9 → S10 via TS5). Here, the value of the energy barrier is higher than that when the first H$_2$ molecule dissociated in the similar way (TS2). This may be due to the weakening of the interaction between the H$_2$ reactant and the N atom, since the latter has already formed an N–H bond. What follows, the aniline product desorbs from the Pt$_2$ system, which is facilitated by a change in the configuration of the hydrogen adsorbate on Pt (S10 → S11). Then, the remaining one hydrogen atom and one hydroxyl group on the Pt$_2$ site form a water molecule through hydrogen transfer (S11 → S12 via TS6), and upon the desorption of water (S12 → S13), the catalyst restores. It may be worth noting that the process from S10 to S13 could take place more easily than what the energy profile reflects, since the transfer of the hydrogen atom will be promoted by the quantum tunneling effect, and the water desorption can be promoted by the competitive adsorption of the nitrobenzene reactant.

As aforementioned, the underlying reason why Pt$_2$ can exhibit the unique catalytic performance in the nitrobenzene hydrogenation, compared with the Pt single atoms and nanoparticles, is that not only the activation and dissociation of the H$_2$ reactant can easily occur, the desorption of the generated aniline product is also a facile process. By contrast, however, the above two features are not simultaneously possessed in either Pt$_1$ or the Pt nanoparticles. We first compare the difference between Pt$_1$ and the dual-atom catalyst. In Fig. 5 and Supplementary Fig. 27, we present the reaction pathway and the computational energy profile of nitrobenzene hydrogenation on Pt$_1$/g-C$_3$N$_4$. Since the Pt$_1$/g-C$_3$N$_4$ system contains only one Pt atom, the unsaturated N and O atoms cannot be generated as in the case of Pt$_2$. Fortunately, since the Pt atom directly participates in the dissociation of the first H$_2$ molecule (S1 → S2 via TS1), the

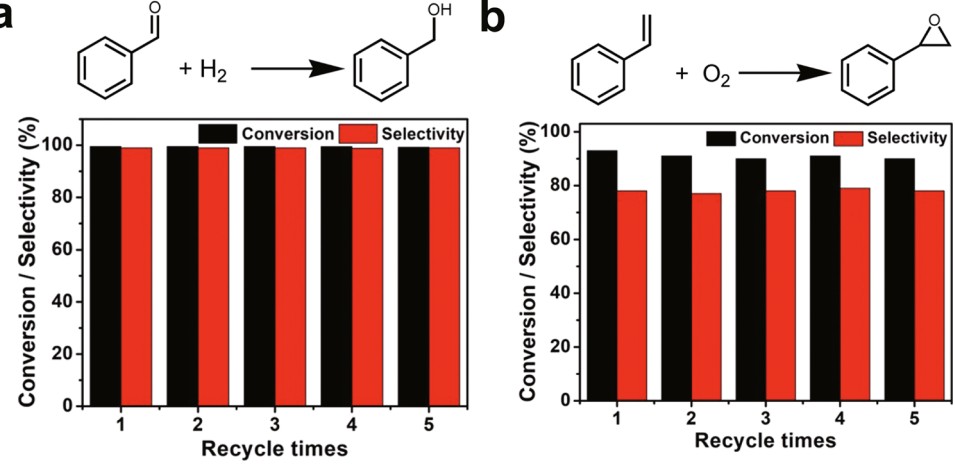

**Fig. 6 Hydrogenation of benzaldehyde and epoxidation of styrene. a** Catalytic performance for the hydrogenation of benzaldehyde by using the Pt$_2$/mpg-C$_3$N$_4$ catalyst. **b** Corresponding catalytic performance for the epoxidation of styrene.

corresponding energy barrier is not very high, being only 0.66 eV. However, since neither Pt atoms nor the two unsaturated N and O atoms are present in the dissociation of the second (S4 → S5 via TS3) and the third (S6 → S7 via TS4) H$_2$ molecules, the reaction energy barriers significantly increase, both of which reach 1.50 eV. Thus, it is not surprising that the Pt$_1$ system cannot exhibit the same excellent catalytic properties as Pt$_2$.

Regarding the Pt nanoparticles, it was found that the hydrogenation was induced and assisted by the adsorbed H atoms that were produced via a spillover process on the outermost layers of the nanoparticles (simulated using a Pt (111) surface in the simulations)[52]. The overall energy barrier was calculated to be only 0.75 eV (ref. [50]), indicating that the occurrence of the hydrogenation is not difficult. The obstacle, however, comes from the desorption step of the aniline product. Our calculations show that the adsorption energy of aniline on Pt (111) is as high as −1.70 eV, and in the adsorption configuration (Supplementary Fig. 28A), several C atoms of the phenyl group are involved in the bonding with the surface Pt atoms. It means that upon desorption, the aniline molecule needs to overcome a high energy barrier. Furthermore, since the adsorption of nitrobenzene (showing an adsorption energy of −1.35 eV) is weaker than that of aniline on Pt(111), the desorption of the aniline product cannot be promoted via a competitive adsorption of the reactant. Such problems, however, do not appear in the Pt$_2$ system. Despite that the adsorption energy of aniline on Pt$_2$ is as high as −2.58 eV, the adsorption of the nitrobenzene reactant is stronger, showing an adsorption energy of −3.16 eV. The stronger interaction of the adsorption site with the nitrobenzene reactant is able to promote aniline desorption. Moreover, there is only one N–Pt bond connecting aniline with the Pt$_2$ catalyst (Supplementary Fig. 28C), and the configuration change of the hydrogen adsorbate on Pt can also promote the desorption process (S10 → S11 in Fig. 4). Thus, the aniline desorption can be easily achieved and is no longer an obstacle on Pt$_2$.

By comparing the reaction pathways in Figs. 4 and 5, one can see that for the Pt$_1$/g-C$_3$N$_4$ system, the g-C$_3$N$_4$ framework not only serves as a substrate to anchor the Pt atoms, but also, its C atoms can directly participate in the nitrobenzene hydrogenation. For example, in the cleavage of the first N–O bond on Pt$_1$/g-C$_3$N$_4$ (Fig. 5), an adjacent carbon atom to Pt acts as the adsorption site to stabilize the produced OH radical (S2 → S3 via TS2). In the activation and dissociation of the third H$_2$ molecule (S6 → S7 via TS4), this carbon atom, together with the oxygen atom nearby, stabilize the two produced hydrogen atoms. The adsorbates on

the carbon atom can also affect the electronic structure of this atom, as well as that of the entire g-C$_3$N$_4$ framework. In the Supplementary Fig. 29, we present the calculated electronic DOS of this C atom (left panel) and the g-C$_3$N$_4$ framework (right panel) for the configurations S2 (before OH connects to the C atom), S3 (with OH bound to the C atom), and S5 (after OH leaves the C atom), as shown in Fig. 5. One can see that when the OH group does not form a bond with the C atom, no matter the system adopts the configuration S2 or S5, the electronic structures of both the C atom and the g-C$_3$N$_4$ framework are not much different. However, when the OH radical is attached to the C atom, the electronic structures undergo obvious changes: for the C atom that is bound to OH, the unoccupied electronic states within 3 eV above the Fermi level disappear; while for the g-C$_3$N$_4$ framework, the entire DOS undergoes a right shift relative to the Fermi level. The changes in the above electronic structures also shows that the OH group has a strong interaction with the C atom, further supporting that OH is a radical rather than an anion.

**Hydrogenation of benzaldehyde and epoxidation of alkenes.** The produced Pt$_2$/mpg-C$_3$N$_4$ catalyst is versatile and can be employed in other important reactions besides the selective hydrogenation of nitrobenzene. For example, under the conditions of 8 MPa of an H$_2$ and N$_2$ mixture (1:1) at 120 °C, the Pt$_2$/mpg-C$_3$N$_4$ sample showed optimal catalytic performance toward the hydrogenation of benzaldehyde to benzyl alcohol (Fig. 6a). Specifically, ˃99% conversion and ˃99% selectivity were achieved for 7 h. Moreover, the catalyst can be reused at least five times without obvious loss of the activity (Fig. 6a). In Supplementary Figs. 30 and 31, we display the reaction pathway and the computational energy profile of benzaldehyde hydrogenation on the Pt$_2$/g-C$_3$N$_4$ catalyst, showing that the Pt$_2$ species can effectively catalyze the hydrogenation of benzaldehyde toward benzyl alcohol. Especially, the H$_2$ dissociation can be promoted by the Pt$_2$ species, and the C=O bond can also be activated by Pt$_2$ (S1), as reflected in the elongation of the corresponding bond length from 1.23 Å (the value of an isolated benzaldehyde molecule) to 1.40 Å. In addition, the desorption of benzyl alcohol can also be promoted by the configuration change of the hydrogen adsorbate (S5 → S6). These are similar to the reaction pattern of the aforementioned nitrobenzene hydrogenation, demonstrating the unique role of the dual-atom species in the catalytic hydrogenation reactions.

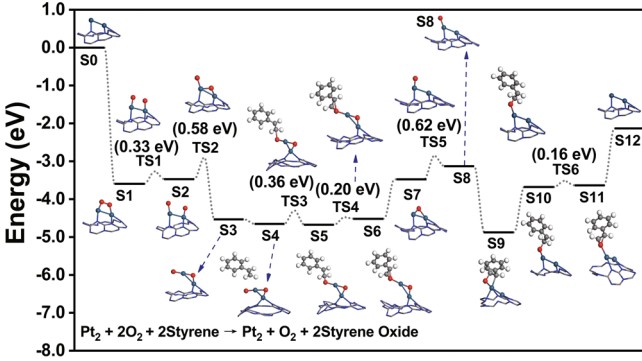

**Fig. 7 First-principles calculations of styrene epoxidation on Pt₂/g-C₃N₄.** Reaction pathway and computational energy profile of styrene epoxidation on the Pt₂/g-C₃N₄ catalyst. The label S0 represents the initial state and the subsequent labels S1–S11 represent a series of intermediate states. The labels TS1–TS6 (TS is short for transition states) represent a series of transition states. Here, only the key structures, i.e., the Pt₂ catalytic system as well as the adsorbate bound on it, are shown. The information regarding reactant molecules which have not been adsorbed, and/or product molecules which have been desorbed are labeled in the Supplementary Information (Supplementary Fig. 32). The teal, gray, blue, red, and white spheres represent the Pt, C, N, O, and H atoms, respectively.

As another type of significantly important reactions[53], the epoxidation of alkenes in liquid relies on extensive use of expensive oxidants or co-reagents, which leads to a large increase in the cost. Our prepared Pt₂/mpg-C₃N₄ catalyst exhibited excellent catalytic performance toward the epoxidation of styrene when only O₂ molecules were used as the oxidant, which well circumvents the above problem. In Fig. 6b, one can see that Pt₂/mpg-C₃N₄ exhibited a conversion of 93% and a selectivity of 78% after 12 h, and this is one of the best results for the epoxidation of styrene[54,55]. Moreover, the Pt₂/mpg-C₃N₄ catalyst can be reused at least five times without obvious loss of the activity (Fig. 6b). To understand the detailed process of the reaction, in Fig. 7 and Supplementary Fig. 32, we display the corresponding pathway and the energy profile of the styrene epoxidation. It should be noted that since the reaction is carried out in an O₂ atmosphere, the active site is no longer the isolated dual Pt atoms as in the cases of the nitrobenzene and benzaldehyde hydrogenations. Here, an O₂ molecule first adsorbs and then transforms into two oxygen atoms by overcoming an energy barrier of 0.58 eV (S0 → S3 via TS1 and TS2). The generated Pt₂O₂ species (S3) contains a one-coordinated oxygen atom and a two-coordinated oxygen atom, consistent with what we have observed in the EXAFS measurements. The Pt₂O₂ species, in fact, is the true active site of the styrene epoxidation. In the next few steps (from S4 to S6), the one-coordinated oxygen atom interacts with the C=C bond of the alkene and converts styrene to the corresponding epoxide. As the first styrene oxide molecule desorbs, there is only a two-coordinated oxygen atom locating at the Pt₂ site (S7). Such type of oxygen can be converted to a new one-coordinated oxygen atom by crossing an energy barrier of 0.62 eV (S7 → S8 via TS5), and then, participates in the next few steps of the epoxidation reaction (from S9 to S11). Upon the desorption of the second styrene oxide molecule, as well as the adsorption of another O₂ molecule, the catalytic reaction cycle starts again. In Fig. 7, one can see that the one-coordinated oxygen atoms can directly participate in the epoxidation reaction, while the two-coordinated oxygen atom is not involved in the reaction until it is converted to the one-coordinated configuration. It is not surprising because the one-coordinated oxygen atom connects with Pt₂ via only one

chemical bond and is thus less bound, which makes it easier to participate in the reaction and be grabbed by the styrene molecules. This phenomenon is very similar to the case of the diatomic Fe₂ system, which we used as dual-atom catalyst in the epoxidation of *trans*-stilbene molecules[28].

## Discussion

In conclusion, we have reported an atomically monodispersed dual-atom Pt heterogeneous catalyst, which exhibited excellent catalytic properties in selective hydrogenation and epoxidation reactions. Multiple characterization techniques, including AC STEM, XAFS spectra, and first-principles simulations, were employed to capture the structural and chemical nature of the dual-atom Pt species. Compared with the mpg-C₃N₄-supported Pt single-atom catalysts and ultra-small Pt nanoparticles (~2 nm), Pt₂/mpg-C₃N₄ exhibited much better performance toward the selective hydrogenation of nitrobenzene, due to the facile dissociation of the H₂ reactant and the effective release of the aniline product. More interestingly, the prepared Pt₂/mpg-C₃N₄ catalyst is versatile and can be applied in catalyzing other important reactions, like the hydrogenation of aldehyde compounds and the epoxidation of alkenes.

## Methods

**Materials.** Cyanamide (98%), LUDOXR AS-40 colloidal silica 40 wt% suspension in H₂O, ammonium hydrogen difluoride, hydrogen hexachloroplatinate(IV) hydrate, platinum(II) 2,4-pentanedionate, and (ethylenediamine)iodoplatinum(II) dimer dinitrate were purchased from Innochem. Ethanol, N,N-dimethylformamide, and n-hexane were purchased from Sinopharm Chemical Reagent Co. Ltd. Oleylamine and borane-*tert*-butylamine were purchased from Sigma-Aldrich Reagent Company.

## Methods

*Synthesis of mpg-C₃N₄, and Pt nanoparticles.* The synthesis of mpg-C₃N₄ was according to the previous method without any change[46]. In the synthesis of the Pt nanoparticles, 20 mg Pt(II) acetylacetonate was dissolved in 10 mL octadecenylamine (OAm) at 120 °C. A solution of 100 mg borane-*tert*-butylamine in 2 mL OAm was added quickly into the above solution. After 2 min, the flask was heated to 140 °C and then kept at 140 °C for an hour. After that, the solution was cooled to room temperature, followed by ethanol washing.

*Synthesis of Pt₂/mpg-C₃N₄, Pt₁/mpg-C₃N₄, and Pt nanoparticles/mpg-C₃N₄.* In the typical synthesis of Pt₂/mpg-C₃N₄, 500 mg mpg-C₃N₄ and 5 mg (ethylenediamine) iodoplatinum(II) dimer dinitrate were dissolved in 100 mL DMF under the stirring condition. After continuous stirring for ~12 h, the resulting product was centrifuged and the supernatant was discarded. The obtained precipitate was washed with DMF and methanol. The as-prepared powder was treated under the N₂ atmosphere at 300 °C for 2 h. The loading of Pt, determined by inductively coupled plasma optical emission spectrometer (ICP-OES) analysis, was 0.15 wt%. Thermogravimetric analysis showed no weight loss at 300 °C, indicating that the ligands were removed completely (Supplementary Fig. 33). The synthesis methods of Pt₁/mpg-C₃N₄ and Pt nanoparticles/mpg-C₃N₄ are similar to that of Pt₂/mpg-C₃N₄ with some modifications. In the synthesis of Pt₁/mpg-C₃N₄, 500 mg mpg-C₃N₄ was dissolved in 100 mL H₂O. Then, a solution of 2.5 mg H₂PtCl₆ in 10 mL H₂O was added to the above solution under vigorous stirring. After continuous stirring for ~12 h, the resulting product was centrifuged and the supernatant was discarded. The as-prepared powder was treated under the N₂ atmosphere at 125 °C for 2 h. The loading of Pt, determined by ICP-OES analysis, is 0.18 wt%. In the synthesis of Pt nanoparticles/mpg-C₃N₄, 5 mg Pt nanoparticles and 500 mg mpg-C₃N₄ were dissolved in a mix solvent of 200 mL ethanol and n-hexane (1:1 v/v) under stirring at room temperature for 12 h. The product was separated by centrifugation, then washed with methanol. The as-prepared powder was treated under the N₂ atmosphere at 125 °C for 2 h. The Pt loading is 0.42 wt% as determined by ICP-OES analysis.

### Catalytic tests

*Typical procedure for the hydrogenation of nitrocompound.* In the typical experiment, the reaction mixture containing 1 mmol nitrocompound (nitrobenzene, p-nitrophenol, p-nitrotoluene, tetrachloro-nitrobenzene, and tetrabromonitrobenzene), catalyst (equal 0.000373 mmol Pt for Pt₁/mpg-C₃N₄, Pt₂/mpg-C₃N₄, and Pt nanoparticles/mpg-C₃N₄ or 50 mg mpg-C₃N₄), and 10 mL isopropanol were loaded into the reactor. The reactor was sealed and pressurized with 1 MPa H₂ and 3 MPa N₂ to a setting point. The reaction was then heated to the 100 °C and kept for 3 h. The products were identified by gas chromatography–mass spectrometer (GC–MS) and gas chromatography (GC).

Following the hydrogenation reaction, the reaction mixture was centrifuged to recover the catalyst, which was first washed with acetone and then water, followed by drying under vacuum oven at 50 °C before being employed for the next catalytic test.

*Typical procedure for the hydrogenation of benzaldehyde.* In the typical experiment, the reaction mixture containing benzaldehyde (1 mmol), 50 mg $Pt_2/mpg-C_3N_4$, and 10 mL isopropanol were loaded into the reactor. The reactor was sealed, purged three times with 1 MPa of $N_2$ at and then pressurized at 8 MPa of an $H_2$ and $N_2$ mixture (1:1) to a setting point. The reaction was then heated to the 120 °C temperature and kept for 9 h.

*Typical procedure for the epoxidation of styrene.* A total of 1 mmol styrene, 20 mg $Pt_2/mpg-C_3N_4$, and 5 mL 1,4-dioxane were mixed in a 20 mL Schleck tube. Then, the air in the Schleck tube was removed by an oil pump. An $O_2$ balloon was used to blow ~1 atm $O_2$ into the tube. Finally, the reaction vessel was heated in an oil bath at 100 °C for 12 h.

The products were identified by the gas chromatography–mass spectrometry (GC–MS, Thermo Fisher Scientific-TXQ Quntum XLS), and were quantitatively analyzed by the GC (Shimadzu, GC-2010 Plus), equipped with the flame ionization detector and a (30 m × 0.25 mm × 0.25 mm) KB-WAX capillary column (Kromat Corporation, USA), using *n*-octanol as the internal standard. Operation parameters for the GC–MS measurements: the inlet temperature was 250 °C, the MS transfer line temperature was 250 °C, and the ion source temperature was 280 °C. The column temperature was first kept at 40 °C for 1 min, and then raised to 150 °C with a ramp rate of 20 °C min$^{-1}$, and was later raised to 250 °C with a ramp rate of 15 °C min$^{-1}$. Finally, it was kept at 250 °C for 7 min. Operation parameters for the GC measurements: the vaporization temperature and the detector temperature were both 270 °C. The column temperature was first kept at 40 °C for 1 min, and then raised to 240 °C with a ramp rate of 10 °C min$^{-1}$ and was finally kept at 240 °C for 6 min.

**Characterization.** XAFS measurements and analysis. The XAFS data of Pt $L_3$-edge was collected in fluorescence excitation mode using a Lytle detector at 1W1B station in Beijing Synchrotron Radiation Facility. The EXAFS data were processed by the standard procedures using the ATHENA module implemented in the IFEFFIT software packages. To obtain the quantitative structural parameters around central atoms, the least-squares curve parameter fitting was performed, using the ARTEMIS module of IFEFFIT software packages.

**Details of calculations.** Most of the calculations were conducted using the Vienna ab initio simulation package[56,57]. The projector augmented wave approach[58] was employed, and the energy cutoff of the plane-wave basis set was set to 500 eV. The exchange–correlation interactions were described by the optPBE-vdW functional[59,60], which explicitly includes effects of van der Waals forces. The first Brillouin zone was sampled using a $3 \times 3 \times 1$ Monkhorst-Pack grid[61]. To simulate the mpg-$C_3N_4$ and the Pt nanoparticles, a g-$C_3N_4$ monolayer and a Pt(111) slab model (four layers with 16 Pt atoms within each layer) were employed, respectively. Structural relaxations were performed until the maximum residual force on each atom was <0.03 eV Å$^{-1}$. Transition states were located using the climbing-image nudged elastic band method[62] with a force criterion of 0.10 eV Å$^{-1}$. Each transition state has been confirmed via the vibrational mode analysis to ensure that it is indeed connected to the correct initial and final states of the elementary reaction step. The corresponding imaginary frequencies were listed in Supplementary Tables 3–6 and the raw file (.xyz) for the associated vibrational modes were uploaded to the NoMaD repository (see the "Data availability" section). The LUMO orbital of the isolated nitrobenzene molecule is calculated with the Gaussian 16 package[63] using the PBE[64] functional and the 6-31 G(d) basis set. All structures were visualized using the program VESTA[65].

**Data availability**

The computational data for Figs. 4, 5, and 7, and Supplementary Fig. 30 is available in the NoMaD Repository (http://nomad-repository.eu/) via https://doi.org/10.17172/NOMAD/2021.04.28-1. And all relevant data that support the findings of this study are available from the authors upon reasonable request.

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

## Acknowledgements

This work was supported by the National Key R&D Program of China (2018YFA0702003), the National Natural Science Foundation of China (21890383, 21671117, 21871159, and 21803036), and Science and Technology Key Project of Guangdong Province of China (2020B010188002). Q.F. also thanks the Shandong Provincial Natural Science Foundation, China (ZR2018QB005) and the Young Scholars Program of Shandong University (2018WLJH49). S.T. thanks supports from the Fundamental Research Funds of Beijing University of Chemical Technology (buctrc202107). First-principles calculations were performed on the HPC Cloud Platform of Shandong University and the super-computing system in Shanghai-SCC. We thank the BL14W1 station in Shanghai Synchrotron Radiation Facility (SSRF) and 1W1B station in Beijing Synchrotron Radiation Facility (BSRF) for XAFS measurement.

## Author contributions

S.T. conducted and designed the experiments, analyzed the data, and wrote the paper. B.W. and Z.H. finished parts of the simulations. W.G. and C.C did the hydrogenation reaction and analyzed the data. Q.X., W.C, Y.Z., J.Y., and X.S helped to analyze the data. Q.Z. and L.G. provided AC STEM techniques. Q.F. conducted the first-principles calculations and wrote the corresponding contents of the paper. Y.B. helped to analyze the calculation results. H.Z. designed the hydrogenation reaction. D.W. and Y.L. conceived and designed the research project. All authors contributed to the preparation of the manuscript.

## Competing interests

The authors declare no competing interests.
