## [Peer Review File · Nature Communications]

REVIEWER COMMENTS

Reviewer #1 (Remarks to the Author):

The manuscript "Dual-atom Pt heterogeneous catalyst with excellent catalytic performances for the selective hydrogenation and epoxidation" by Wang, Fu et al. deals with the synthesis, the characterization and the catalytic application of the Pt₂/mpg-C₃N₄ material. The latter point has been investigated through a computational investigation. According to the present reviewer, the experimental part is a routine characterization in line with other already reported characterization of functionalized materials with simple metal atoms on top. The manuscript does not provide any reasonable explanation of why the dual-atom is more efficient than the single metal atom or Pt nanoparticles. In this regards, the computational analysis, which must be fundamental in explaining the reactivity, only provides energy numerical variations or energy barriers. In addition, in the catalytic cycle the authors suggests the intermediate S₈, with a hydride on the Pt. How does the catalyst restore? According to the present reviewer, the computational investigation has to be completely reviewed by introducing the calculations with single Pt atom as well as also the investigation of the epoxidation catalytic process, which is not reported in the manuscript. A reasonable explanation of the different reactivities between single and dual Pt atom has to be provided.

On these basis, the manuscript is not suitable for a top high level of journal such as Nature Communications and, also before submission on other journal, the manuscript needs to be improved and written in the form of full paper rather than Communication.

Reviewer #2 (Remarks to the Author):

This contribution presents new catalysts composed of two Pt atoms linked and monodispersed on a mesoporous graphitic carbon nitride support. As the authors claim, the preparation of this kind of catalysts is challenging, but it can open the way to new possibilities due to the different reactivity and mechanisms that can exhibit.

- When the authors talk about ultra-small nanoparticles (page 3), they should define below which size they are referring to

- In the results section: Synthesis and characterization of Pt₂/mpg-C₃N₄ samples in page 3, when writing about the synthesis, the authors should add: (See Experimental section for more details) so the reader is directed to the right section where he/she will be able to find more details

- In page 4, the authors write: "After the pyrolysis procedure, no infrared spectroscopy absorption peaks that correspond to the ligands of the precursor were observed in the Pt₂/mpg-C₃N₄ sample, which supported a complete removal of the ligand molecules (Supplementary Fig. S4)."

The authors should describe better which are the absorption contributions of the ligands in the main text or at least in the suppl. info.

- In page 4, it is written: "Besides, a few isolated bright dots (marked with green circles) were also observed, which we attributed to an overlap of paired Pt dots in the incident beam direction or an incomplete imaging of the dual-atom Pt species due to incomplete focusing (Supplementary Fig. 1c)"

In this case, the authors should provide evidence that a better focusing allows a better visualization of the Pt-Pt paired atoms. It would be nice to show in the Suppl. Info a figure with the same "problem" and then how when staying in the same region and changing the focus, the structures are better defined supporting what the authors claim. This is critical since it will justify the more important insight from this contribution, which is the possibility to synthesize those structures monodispersed on the substrate, which is quite challenging and opens new possibilities

in heterogeneous catalysis

- When referring to some figures in the main text the authors write. For example in page 4 it can be found twice: "(Supplementary Fig. 1c). ", but the figure of interest belongs to the main text so please, remove "Supplementary" in order to avoid misleading the reader. This also can be found in other cases through the manuscript.

- Page 6: "The configuration of the Pt₂/mpg-C₃N₄ sample was also explored by extensive first-principles calculations, with the optimized geometry shown in the inset of Figure 2c."

There is no inset figure in 2c, please correct this. Later on, in the same page, the authors write again: "Fortunately, the Pt-N/O and the Pt-Pt bond lengths were calculated to be 1.96±0.14 Å and 2.55 Å, with the corresponding coordination numbers being 2.5 and 1.0 based on the model in Figure 2c."

And the model is not in Fig. 2c

- At the end of page 6, beginning of page 7, the authors write: "After five cycles, the HAADF-STEM image and EXAFS spectrum of the Pt₂/mpg-C₃N₄ material did not exhibit any change, indicating that the Pt species were still well dispersed as dual-atom Pt pairs (Supplementary Fig. S13-14)."

Considering Fig. S13, this statement, in my opinion, is not clear. Authors should show aberration-corrected (AC) HAADF-STEM images as in Figs. 1c-d, marking as in these figures, the species of interest. This is critical since the authors must demonstrate that there are no structural changes under reaction conditions, that could hinder the application of these catalyst candidates.

- Page 7: "we have explored the hydrogenation of several other nitroarene derivatives, including p-nitrophenol, p-nitrotoluene, tetrachloro-nitrobenzene, and tetrabromonitrobenzene. We found that Pt₂/mpg-C₃N₄ exhibits excellent yields for all corresponding anilines (Fig. 3c)."
(...) More info required in the Suppl. Info!

- In page 8, the authors write: "As we have expected based on the elongation of the N-O bonds, the N-O rupture is easy to occur (S2)"

Define S2 at least the first time you use it and refer to the Fig. 5a (please, define different parts of the figure with letters as done in other figures (a and b)). I would start commenting on the starting state (S1) which corresponds to the nitrobenzene already adsorbed onto the sample of interest. Could the authors comment on the initial adsorption stage from an energetic point of view?

- Following the previous text, in the same page: "during which a barrier of only 0.60 eV (TS1) "
Define TS the first time is used

- At the end of page 8 and beginning of page 9: "Here, the one-coordinated oxygen atom on the Pt₂ species (Fig. 2f) plays an important role in the catalytic reaction, which is reminiscent of the diatomic Fe₂ system in catalyzing the epoxidation of trans-stilbene."²
The authors should discuss the role or absence of role of the two-coordinated O atom

- In the section Catalytic test in page 11, the authors write:
"The reactor was sealed then pressurized with 1 MPa H₂ and 3 MPa N₂ to a setting point. " while in the main text in page 6, they write: "Under the conditions of 4 MPa H₂ pressure and 100°C, a conversion of >99% was obtained on the Pt₂/mpg-C₃N₄ catalyst for the hydrogenation of nitrobenzene to aniline"
(...)
Please could you clarify which are the reaction conditions?

- Following the same issue, in the following section: "Typical procedure for the hydrogenation of

benzaldehyde", it is written: "The reactor was sealed, purged three times with N₂ at 1 MPa and then pressurized with 4 MPa H₂ and 4 MPa N₂ to a setting point."

If the total pressure under reduction conditions is 8 MPa, they should write this in a more clear way, e.g. and then pressurized at 8 MPa of an H₂ and N₂ mixture (1:1) to a setting point

- Page 11: "The products were identified by GC-MS and GC." The authors should give more details on the GC-MS and GC instruments used and also about the experimental conditions applied for these measurements.

- Page 20, Fig. 1d: Could the authors explain features like the signaled in the attached picture? To get 100% Pt₂ structures or 100% Pt atoms supported sounds impossible and I miss more discussion about what the authors have found when studying the samples prepared by electron microscopy about the possibility to get also different structures in the different synthesis followed.

- Page 21, Fig. 2c. As I already said before, in the figure caption the authors mention: "The inset of c is the optimized geometry of Pt₂/mpg-C₃N₄." and there is no inset.

- Page 24, Fig. 4: The distances in b are difficult to see, please improve. Furthermore, I would suggest to use a different color for the Pt atoms

As a general comment, the characterization by transmission electron microscopy should be reinforced. The dual Pt structures are clearly shown, as expected, in the aberration corrected measurements but not in the other. Therefore, these measurements should also be carried out in the post-mortem catalysts. Furthermore, in situ XAS measurements would be appreciated and would give a plus to this work. Notwithstanding, the synthesis and the control of the dispersion of the catalytic entities is a big asset. As stated before, the characterization of the catalysts by AC-TEM measurements after the reaction in order to demonstrate that the dispersion is kept and there is no sintering is of key importance.

SUPPL. INFO:

- page 1: Tian et al instead of tian ...

- page 9: In the figure S8 caption, give a brief explanation of the inset as it was done in Fig. 2c caption (although in this case, the comment should be removed or the inset added since it is not displayed!)

-page 14, Fig. S12: Can you show the images before and after the reaction with the same scale? You should use the same scale also in Fig. S3 and it would be nice for the reader to see a direct comparison of both images, from before and after the reaction.

-page 15, Fig. S13, following my previous comment...you should show two pictures with the same scale, one before and one after the reaction to allow for a direct comparison.

-page 17, Fig. S15, I would recommend to use a different color for the Pt atoms in order to get a better contrast

Apart from these comments/questions/suggestions, there are other typos to consider:

- page 6, 100°C, 100 °C instead

- page 8: 120°C, 120 °C instead

- page 25, figure 6 caption: hhydrogenation

- Although in the main text the authors have done a good job describing in a clear and concise way the different parts, in the section Materials and methods there are several typos and grammatical issues to fix: "In preparation of Pt nanoparticles, ", "for a further 1 hour", "showed almost no more weight was losing", "was dissolved in (the) 100 mL H₂O", "in a 20 ml of Schleck tube", "was heated in (a) oil", etc

Reviewer #3 (Remarks to the Author):

The authors reported dual-atom Pt heterogeneous catalyst supported on mpg-C₃N₄ with excellent catalytic performances for the selective hydrogenation and epoxidation. The topic is very interesting. I am not an expert from experiment, but I can see the experimental synthesis and characterization have been carried out with full care. The dual-atom Pt/mpg-C₃N₄ catalyst has demonstrated the highest catalytic performance compared to single Pt/C₃N₄ atom catalyst. Overall the paper is well organized. I would like to recommend it publishing in Nature Communications after addressing the following minor issues.

(i) It would be good for authors to give more comparison between Pt₁, Pt₂ and Pt nanoparticles and provide a more clear picture on why the Pt₂/mpg-C₃N₄ catalyst can achieve high catalysis performance.

(ii) Can authors provide more recycling times for the Pt₂/mpg-C₃N₄ catalyst?

(iii) The distortion of Pt₂/mpg-C₃N₄ is highly expected. Can authors give some comments on it?

(iv) Some recent theoretical works on the development of Pt/g-C₃N₄ based catalysts for the hydrogenation and reduction reactions could be cited [e.g. JACS 138(2016)6292; Nano Research 12 (2019) 1817].

Responses to the Reviewers' Comments

We are very grateful to all the reviewers for their insightful comments and valuable suggestions, which have helped us a lot to improve the quality of the manuscript. In the revision, we have performed additional experiments and added many new simulation results, trying our best to fully address the comments and suggestions. Below please find our point-by-point responses.

Reply to the Report of Reviewer 1

Referee: *The manuscript “Dual-atom Pt heterogeneous catalyst with excellent catalytic performances for the selective hydrogenation and epoxidation” by Wang, Fu et al. deals with the synthesis, the characterization and the catalytic application of the Pt₂/mpg-C₃N₄ material. The latter point has been investigated through a computational investigation.*

Reply: We thank the Referee for his/her carefully reading, insightful comments, and valuable suggestions. After having addressed the Referee's comments, the quality of the manuscript has been significantly improved.

Referee: *According to the present reviewer, the experimental part is a routine characterization in line with other already reported characterization of functionalized materials with simple metal atoms on top.*

Reply: Besides the experiments that had been reported in the original manuscript, we have further performed time-dependent XAFS measurements to detect the changes in the oxidation state and chemical bonding of the Pt₂ species during the reaction process. The results have been added in the Supplementary Figs. S21-22 in the revised version. Here, the Pt L₃-edge in the XANES spectra of Pt₂/mpg-C₃N₄ showed that the intensity of the white line peaks became lower during the reaction (Supplementary Fig. S21), meaning that the oxidation state of Pt was smaller than that in the initial state. It is not surprising because the oxygen atoms attached to Pt can be removed by the hydrogen molecules. Besides, the EXAFS spectra showed that the first shell peak shifted from 1.57 Å to 1.55 Å (Supplementary Fig. S22), indicating that shorter chemical bonds like that of Pt-H appeared in the reaction process. This information has been added on Page 8 in the revised manuscript.

Fig. S21. Time-dependent XANES spectra of Pt₂/mpg-C₃N₄ during the hydrogenation of nitrobenzene to aniline.

Fig. S22. Time-dependent EXAFS spectra of Pt₂/mpg-C₃N₄ during the hydrogenation of nitrobenzene to aniline.

Referee: *The manuscript does not provide any reasonable explanation of why the dual-atom is more efficient than the single metal atom or Pt nanoparticles. In this regards, the computational analysis, which must be fundamental in explaining the reactivity, only provides energy numerical variations or energy barriers.*

Reply: We agree with and appreciate the Referee’s insightful comments, which prompt us to explore the underlying reason for the unique and excellent catalytic properties of the dual-atom Pt₂ system. According to our analysis of new simulation results, the reason why Pt₂ is more efficient than the single atom or Pt nanoparticles in the nitrobenzene hydrogenation is that, not only the N-O bonds of nitrobenzene can be easily broken (assisted by the two Pt atoms) and hydrogenated on Pt₂, the desorption of the generated aniline product is also a facile process (promoted by the configuration change of a hydrogen adsorbate and the competitive adsorption of the nitrobenzene reactant), which helps the restart of the catalytic cycle. By contrast, however, the above two features are not simultaneously possessed in either Pt₁ or the

Pt nanoparticles.

We first compare the difference between Pt_1 and the dual-atom catalyst. The reaction pathways and the computational energy profiles of the nitrobenzene hydrogenation on $\text{Pt}_2/\text{g-C}_3\text{N}_4$ and $\text{Pt}_1/\text{g-C}_3\text{N}_4$ are displayed in Fig. 4 and Fig. 5, respectively. Here, since the active site of the Pt_1 system contains only one Pt atom, the occurrence of the N-O bond cleavage is not as easy as that on Pt_2 , because at least two adsorption sites are required to stabilize the produced groups after the bond rupture ($\text{S1} \rightarrow \text{S2}$ via TS1 and $\text{S7} \rightarrow \text{S8}$ via TS4 in Fig. 4). Although the breaking of the N-O bond can also be assisted by the Pt atom and an adjacent C atom on the Pt_1 system, as shown in the $\text{S2} \rightarrow \text{S3}$ process via TS2 in Fig. 5, the presence of such C atoms cannot *always* be maintained during the reaction process. In the breaking of the second N-O bond ($\text{S6} \rightarrow \text{S7}$ via TS4 in Fig. 5), for example, there is no neighboring C atom involved in and thereby, the corresponding energy barrier becomes as high as 2.31 eV. Thus, it is not surprising that the Pt_1 system cannot exhibit the same excellent catalytic properties as Pt_2 .

Fig. 4. Reaction pathway and computational energy profile of nitrobenzene hydrogenation on the $\text{Pt}_2/\text{g-C}_3\text{N}_4$ catalyst. The label S0 represents the initial state and the subsequent labels S1 – S13 represent a series of intermediate states. The labels TS1 – TS6 represent a series of transition states. Here, only key structures are shown and more details of the compositions and geometries are placed in the supplementary information (Supplementary Fig. S23). The teal, gray, blue, red, and white spheres represent the Pt, C, N, O, and H atoms, respectively.

Fig 5. Reaction pathway and computational energy profile of nitrobenzene hydrogenation on the $\text{Pt}_1/\text{g-C}_3\text{N}_4$ system. The label S0 represents the initial state and the subsequent labels S1 –S10 represent a series of intermediate states. The labels TS1 – TS5 represent a series of transition states. Here, only key structures are shown and more details of the compositions and geometries are placed in the supplementary information (Supplementary Fig. S24). The teal, gray, blue, red, and white spheres represent the Pt, C, N, O, and H atoms, respectively.

On the outermost layer of the Pt nanoparticles (simulated by a Pt(111) surface in the calculations), the hydrogenation is induced and assisted by the adsorbed H atoms that are produced via a spillover process. The overall energy barrier was calculated to be only 0.75 eV [T. Sheng *et al.*, *Chem. Eng. J.* 293, 337-344 (2016)], indicating that the occurrence of the hydrogenation is not difficult. The obstacle, however, comes from the desorption step of the aniline product. Our calculations show that the adsorption energy of aniline on Pt(111) is as high as -1.70 eV, and in the adsorption configuration, several C atoms of the phenyl group are involved in the bonding with the surface Pt atoms (Supplementary Fig. S25A). It means that upon aniline desorption, the product molecule needs to overcome a high energy barrier. Besides, since the adsorption of nitrobenzene (showing an adsorption energy of -1.35 eV) is weaker than that of aniline on Pt(111), the desorption of the aniline product cannot be promoted via the competitive adsorption of the reactant. Such problems, however, do not appear in the Pt_2 system. Despite that the adsorption energy of aniline on Pt_2 is as high as -2.58 eV, the adsorption of the nitrobenzene reactant is stronger, showing an adsorption energy of -3.16 eV. Moreover, in this case, there is only one N-Pt bond connecting aniline with the Pt_2 catalyst (Supplementary Fig. S25C), and the aniline desorption can also be facilitated by a change in the configurations of the hydrogen adsorbate on Pt_2 (S10 \rightarrow S11 in Fig. 4). Thus, the aniline desorption can be easily achieved and is no longer an obstacle on Pt_2 .

Fig. S25. Adsorption configurations of an aniline molecule (A and C) and a nitrobenzene molecule (B and D) on the Pt(111) surface (top) and the Pt₂/g-C₃N₄ catalyst (bottom). The corresponding adsorption energy values are displayed in the respective lower right corners. The teal, gray, blue, red, and white spheres represent the Pt, C, N, O, and H atoms, respectively.

We have added the discussion (on pages 8-10 in the main text), Figs. 4-5 (in the main text) and Fig. S25 (in the supplementary information) in the revised version.

Referee: *In addition, in the catalytic cycle the authors suggest the intermediate S8, with a hydride on the Pt. How does the catalyst restore?*

Reply: We agree with and appreciate the valuable comments of the Referee. In Fig. 4, we have displayed the entire process of the nitrobenzene hydrogenation on Pt₂, which starts from the adsorption of the nitrobenzene reactant, and ends with the desorption of the aniline product and the regeneration of the Pt₂ active site. In the new energy profile, the reaction intermediates (S1 – S13) and the transition states (TS1 – TS6) have been renumbered. After the formation of the intermediate state S10 (numbered as S8 in the original version), the aniline product desorbs from the Pt₂ system, which is facilitated by a change in the configurations of the hydrogen adsorbate on Pt (S10 → S11 in Fig. 4). Then, the remaining one hydrogen atom and one hydroxyl group on the Pt₂ site form a water molecule through hydrogen transfer (S11 → S12 via TS6), and upon the desorption of water (S12 → S13 in Fig. 4), the catalyst restores. It may be worth noting that the above process could take place more easily than what the energy profile reflects, since the transfer of the hydrogen atom will be promoted by the quantum tunneling effect, and the desorption of water can be promoted by the competitive adsorption of the nitrobenzene reactant.

We have added the information on pages 8-9 in the revised version.

Referee: According to the present reviewer, the computational investigation has to completely reviewed by introducing the calculations with single Pt atom as well as also the investigation of the epoxidation catalytic process, which is not reported in the manuscript.

Reply: We agree with and appreciate the Referee's very valuable comments. In this revision, we have carried out more calculations and added many new simulation results in the manuscript, such as the entire process of nitrobenzene hydrogenation on Pt₁ (Fig. 5), the processes of hydroxyl elimination in the nitrobenzene hydrogenation on Pt₂ (S4 → S7 and S10 → S13 in Fig. 4), and the entire process of styrene epoxidation on Pt₂ (Fig. 7). Besides, although it was not explicitly mentioned, we have also calculated and supplemented the reaction pathway and the energy profile for the benzaldehyde hydrogenation on Pt₂ (Supplementary Fig. S26).

Fig. 7. Reaction pathway and computational energy profile of styrene epoxidation on the Pt₂/g-C₃N₄ catalyst. The label S0 represents the initial state and the subsequent labels S1 – S9 represent a series of intermediate states. The labels TS1 – TS5 represent a series of transition states. Here, only key structures are shown and more details of the compositions and geometries are placed in the supplementary information (Supplementary Fig. S28). The teal, gray, blue, red, and white spheres represent the Pt, C, N, O, and H atoms, respectively.

Fig. S26. Reaction pathway and computational energy profile of the hydrogenation of benzaldehyde on the $\text{Pt}_2/\text{g-C}_3\text{N}_4$ catalyst. The label S0 represents the initial state and the subsequent labels S1 – S6 represent a series of intermediate states. The labels TS1 and TS2 represent a series of transition states. Here, only key structures are shown and more details of the compositions and geometries are placed in Supplementary Fig. S27. The teal, gray, blue, red, and white spheres represent the Pt, C, N, O, and H atoms, respectively.

In the revised version, the above contents have been added as Fig. 4, Fig. 5 and Fig. 7 in the main text and as Supplementary Fig. S26 in the Supplementary Information. The corresponding words have also been completed re-written (on pages 8-13).

Referee: *A reasonable explanation of the different reactivities between single and dual Pt atom has to be provided.*

Reply: We agree with and appreciate the Referee's insightful comments. As described in detail in the aforementioned response, the underlying reason why Pt_2 exhibits unique catalytic properties is that, not only the N-O bonds of nitrobenzene can be easily broken (assisted by the two Pt atoms) and hydrogenated on Pt_2 , the desorption of the generated aniline product is also a facile process (promoted by the configuration change of the hydrogen adsorbate and the competitive adsorption of the nitrobenzene reactant). Regarding the Pt_1 system, the occurrence of the N-O bond cleavage is not as easy as that on the Pt_2 system, because at least two adsorption sites are required to stabilize the produced groups after the bond rupture, but now, there is only one Pt atom. Without such two sites, the energy barrier can be up to 2.31 eV, which makes Pt_1 exhibit inferior catalytic performance than Pt_2 .

Referee: *On these basis, the manuscript is not suitable for a top high level of journal such as Nature Communications and, also before submission on other journal, the manuscript needs to be improved and written in the form of full paper rather than Communication.*

Reply: We regret that the original version did not meet the criteria approved by the Referee. In the revision, we have carried out more calculations and added many new simulation results aforementioned. From these calculations, we have gained a deep understanding regarding the underlying reason why Pt₂ is more efficient than the Pt single atom or Pt nanoparticles in the nitrobenzene hydrogenation. Besides, we have learned more about the roles played by the Pt₂ species in the processes of styrene epoxidation and benzaldehyde hydrogenation. Furthermore, we have also performed time-dependent XAFS measurements to detect the changes in the oxidation state and chemical bonding of the Pt₂ species during the reaction process. We are pleased to see that after having addressed the Referee's comments, the quality of the manuscript has been significantly improved. With these changes, we sincerely hope that the Referee finds our revised manuscript suitable for publication in Nature Communications.

Reply to the Report of Reviewer 2

Referee: *This contribution presents new catalysts composed of two Pt atoms linked and monodispersed on a mesoporous graphitic carbon nitride support. As the authors claim, the preparation of this kind of catalysts is challenging, but it can open the way to new possibilities due to the different reactivity and mechanisms that can exhibit.*

Reply: We thank the Referee for his/her carefully reading, positive evaluation, insightful comments, and valuable suggestions. After having addressed the Referee's comments, the quality of the manuscript has been significantly improved.

Referee: *When the authors talk about ultra-small nanoparticles (page 3), they should define below which size they are referring to.*

Reply: We thank the Referee for the kind advice. In the revised version on Page 3 as well in the Abstract and conclusion, we have explicitly mentioned that the size of the Pt nanoparticles is around 2 nm.

Referee: *In the results section: Synthesis and characterization of Pt₂/mpg-C₃N₄ samples in page 3, when writing about the synthesis, the authors should add: (See Experimental section for more details) so the reader is directed to the right section where he/she will be able to find more details.*

Reply: We thank the Referee for the kind advice. In the revised version, we have added "(See Materials and Methods section for more details)" [this section starts from page 13] at the end of the "Synthesis and characterization of Pt₂/mpg-C₃N₄ samples" section on page 4.

Referee: *In page 4, the authors write: "After the pyrolysis procedure, no infrared spectroscopy absorption peaks that correspond to the ligands of the precursor were observed in the Pt₂/mpg-C₃N₄ sample, which supported a complete removal of the ligand molecules (Supplementary Fig. S4)." The authors should describe better which are the absorption contributions of the ligands in the main text or at least in the suppl. info.*

Reply: We thank the Referee for the kind advice. In the revised version on page 4, we have explicitly mentioned the adsorption contributions of the ligands as follows: "the infrared absorption peaks that correspond to the ligands of the precursor (at ~580, 825, 1005, 1050, 1340, 3090, and 3230 cm⁻¹) were not observed". The corresponding IR spectra has been added in the supplementary information as Fig. S4.

Fig. S4. IR patterns of the mpg-C₃N₄ (red), Pt₂ precursor (blue), and Pt₂/mpg-C₃N₄ (black).

Referee: In page 4, it is written: "Besides, a few isolated bright dots (marked with green circles) were also observed, which we attributed to an overlap of paired Pt dots in the incident beam direction or an incomplete imaging of the dual-atom Pt species due to incomplete focusing (Fig. 1c)". In this case, the authors should provide evidence that a better focusing allows a better visualization of the Pt-Pt paired atoms. It would be nice to show in the Suppl. Info a figure with the same "problem" and then how when staying in the same region and changing the focus, the structures are better defined supporting what the authors claim. This is critical since it will justify the more important insight from this contribution, which is the possibility to synthesize those structures monodispersed on the substrate, which is quite challenging and opens new possibilities in heterogeneous catalysis.

Reply: We agree with and appreciate the Referee's very valuable suggestions. In the revision, we have reinforced the AC HAADF-STEM characterization. To confirm the existence and the consequence of the incomplete focusing "problem", we have compared the different images that were collected within the same region of the sample but with the focus of the imaging constantly changed (Supplementary Fig. S6). It can be seen that under different focusing conditions, isolated dots can indeed be imaged as paired dots. For example, with the focusing condition changed, the two isolated bright dots in the areas marked as "1" and "2" in the Supplementary Fig. S6a were imaged as paired bright dots in the Supplementary Figs. S6b and S6c. Meanwhile, the area "3" where an isolated dot was imaged in the Supplementary Figs. S6b and S6c can also exhibit paired bright dots in the Supplementary Fig. S6a. Considering that the Pt₁/mpg-C₃N₄ sample hardly exhibited and cannot maintain the feature of paired bright dots, the above results clearly demonstrate that the reported images in Fig. 1c actually come from the dual-atom Pt species.

Fig. S6. AC HAADF-STEM images of $\text{Pt}_2/\text{mpg-C}_3\text{N}_4$ in the different focusing.

In addition, as a supplement to the “paired bright dots” images: depending on the orientation of the Pt-Pt bond relative to that of the incident beam direction, the appearance of the paired bright dots can be different from each other (Supplementary Fig. S5), because the AC HAADF-STEM imaging only represents a two-dimensional projection of the three-dimensional $\text{Pt}_2/\text{mpg-C}_3\text{N}_4$ samples. [Aydin, C. *et al.*, *Angew. Chem., Int. Ed.* **52**, 5262-5265 (2013); Li, Z. Y. *et al.* *Nature* **451**, 46-48 (2008)].

Fig. S5. (a) AC HAADF-STEM image of $\text{Pt}_2/\text{mpg-C}_3\text{N}_4$. (b) The corresponding proposed orientations of the sample for the different detailed features of Pt_2 in the areas 1, 2, 3, and 4.

In the revised version, we have added the above information on pages 4 and 5 in the main text and on pages 6 and 7 in the supplementary information.

Referee: When referring to some figures in the main text the authors write. For example in page 4 it can be found twice: “(Supplementary Fig. 1c).”, but the figure of interest belongs to the main text so please, remove “Supplementary” in order to avoid misleading the reader. This also can be found in other cases trough the manuscript.

Reply: We thank the Referee for pointing this out. We feel sorry for the mistakes and have corrected them in the revised version.

Referee: Page 6: “The configuration of the $\text{Pt}_2/\text{mpg-C}_3\text{N}_4$ sample was also explored by extensive first-principles calculations, with the optimized geometry shown in the

inset of Figure 2c." There is no inset figure in 2c, please correct this. Later on, in the same page, the authors write again: "Fortunately, the Pt-N/O and the Pt-Pt bond lengths were calculated to be $1.96 \pm 0.14 \text{ \AA}$ and 2.55 \AA , with the corresponding coordination numbers being 2.5 and 1.0 based on the model in Figure 2c." And the model is not in Fig. 2c.

Reply: We thank the Referee for pointing this out. The corresponding figure is in Fig. 2f. We feel sorry for the mistake and have corrected them in the revised version.

Referee: *At the end of page 6, beginning of page 7, the authors write: "After five cycles, the HAADF-STEM image and EXAFS spectrum of the $\text{Pt}_2/\text{mpg-C}_3\text{N}_4$ material did not exhibit any change, indicating that the Pt species were still well dispersed as dual-atom Pt pairs (Supplementary Fig. S13-14)." Considering Fig. S13, this statement, in my opinion, is not clear. Authors should show aberration-corrected (AC) HAADF-STEM images as in Figs. 1c-d, marking as in these figures, the species of interest. This is critical since the authors must demonstrate that there are no structural changes under reaction conditions that could hinder the application of these catalyst candidates.*

Reply: We agree with and appreciate the Referee's very valuable suggestions. In the revised version, we have replaced the original HAADF-STEM figure by a new one (Supplementary Fig. S17 in the revised version), showing the AC HAADF-STEM image of $\text{Pt}_2/\text{mpg-C}_3\text{N}_4$ after the reaction. It can be seen that the paired bright dots remained after 10 cycles (it may be worth noting that the recycling time has also been extended from 5 cycles to 10 cycles in the revision).

Fig. S17. AC HAADF-STEM image of $\text{Pt}_2/\text{mpg-C}_3\text{N}_4$ after the reaction.

Referee: *Page 7: "we have explored the hydrogenation of several other nitroarene derivatives, including p-nitrophenol, p-nitrotoluene, tetrachloro-nitrobenzene, and tetrabromonitrobenzene. We found that $\text{Pt}_2/\text{mpg-C}_3\text{N}_4$ exhibits excellent yields for all corresponding anilines (Fig. 3c)." (...) More info required in the Suppl. Info!*

Reply: We agree with and appreciate the Referee's very valuable suggestions. In the

revised version, we have added some details about the reactions in the Supplementary Table S2.

Table S2. Details about the hydrogenation reactions of nitrobenzene, p-nitrophenol, p-nitrotoluene, tetrachloro-nitrobenzene, and tetrabromonitrobenzene catalyzed by the Pt₂/mpg-C₃N₄ sample.

Catalyst	Substrate	Temperature (□)	Time (h)	Selectivity (%)	Yield (%)
Pt ₂ /mpg-C ₃ N ₄	nitrobenzene	100	4	100	100
Pt ₂ /mpg-C ₃ N ₄	p-nitrophenol	100	4	100	100
Pt ₂ /mpg-C ₃ N ₄	p-nitrotoluene	100	4	100	100
Pt ₂ /mpg-C ₃ N ₄	tetrachloro-nitrobenzene	100	4	100	93
Pt ₂ /mpg-C ₃ N ₄	tetrabromonitrobenzene	100	4	100	96

Standard reaction conditions: nitrocompound (1 mmol), catalyst: Pt₂/mpg-C₃N₄ (equal 0.000373 mmol Pt), isopropanol (10.0 mL) as solvent, $T = 100$ □, $t = 4$ h. Determined by gas chromatography (GC) analysis with n-octanol as internal standard.

The following information has also been added on Page 16 in the revised manuscript: “The products were identified by the gas chromatography–mass spectrometry (GC-MS, Thermo Fisher Scientific-TXQ Quntum XLS), and were quantitatively analyzed by the gas chromatography (GC, Shimadzu, GC-2010 Plus), equipped with the flame ionization detector and a (30 m × 0.25 mm × 0.25 mm) KB-WAX capillary column (Kromat Corporation, USA) using n-octanol as the internal standard. Operation parameters for the GC-MS measurements: the inlet temperature was 250 □, the MS transfer line temperature was 250 □, and the ion source temperature was 280 □. The column temperature was first kept at 40 □ for 1 min, and then raised to 150 □ with a ramp rate of 20 □/min, and was later raised to 250 □ with a ramp rate of 15 □/min. Finally, it was kept at 250 □ for 7 min. Operation parameters for the GC measurements: The vaporization temperature and the detector temperature were both 270 □. The column temperature was first kept at 40 □ for 1 min, and then raised to 240 □ with a ramp rate of 10 □/min and was finally kept at 240 □ for 6 min.”

Referee: In page 8, the authors write: "As we have expected based on the elongation of the N-O bonds, the N-O rupture is easy to occur (S2)". Define S2 at least the first time you use it and refer to the Fig. 5a (please, define different parts of the figure with letters as done in other figures (a and b). I would start commenting on the starting state (S1) which corresponds to the nitrobenzene already adsorbed onto the sample of interest. Could the authors comment on the initial adsorption stage from an energetic point of view?

Reply: We appreciate the Referee’s valuable comments. In the revised version, we have carried out more calculations, added many new simulation results, completely rewritten the corresponding paragraphs (on pages 8-13 in the main text), and

completely redrawn the figures for the reaction pathways, such as the process of nitrobenzene hydrogenation on Pt_2 (Fig. 4 in the main text), the process of nitrobenzene hydrogenation on Pt_1 (Fig. 5 in the main text), the process of styrene epoxidation on Pt_2 (Fig. 7 in the main text), and the process of benzaldehyde hydrogenation on Pt_2 (Fig. S26 in the supplementary information). For these figures, we have added notes similar to the following words in the corresponding captions: “The label S0 represents the initial state and the subsequent labels S1 – S13 represent a series of intermediate states. The labels TS1 – TS6 represent a series of transition states. Here, only key structures are shown and more details of the compositions and geometries are placed in the supplementary information (Supplementary Fig. S23). The teal, gray, blue, red, and white spheres represent the Pt, C, N, O, and H atoms, respectively.”

In Fig. 4, we present the reaction pathway and the computational energy profile of nitrobenzene hydrogenation on the $\text{Pt}_2/\text{g-C}_3\text{N}_4$ catalyst. Here, the initial adsorption of nitrobenzene is shown in the $\text{S0} \rightarrow \text{S1}$ process. The corresponding adsorption energy is calculated to be -3.16 eV (a negative value means exothermic adsorption).

Fig. 4. Reaction pathway and computational energy profile of nitrobenzene hydrogenation on the $\text{Pt}_2/\text{g-C}_3\text{N}_4$ catalyst. The label S0 represents the initial state and the subsequent labels S1 – S13 represent a series of intermediate states. The labels TS1 – TS6 represent a series of transition states. Here, only key structures are shown and more details of the compositions and geometries are placed in the supplementary information (Supplementary Fig. S23). The teal, gray, blue, red, and white spheres represent the Pt, C, N, O, and H atoms, respectively.

Referee: Following the previous text, in the same page: "during which a barrier of only 0.60 eV (TS1) ".

Define TS the first time is used.

Reply: We appreciate the Referee’s valuable comments. As mentioned in the previous

response, the label TS has been defined in the figure captions. The energy barrier has been updated to 0.58 eV after considering dipole corrections to the DFT energies.

Referee: *At the end of page 8 and beginning of page 9: "Here, the one-coordinated oxygen atom on the Pt₂ species (Fig. 2f) plays an important role in the catalytic reaction, which is reminiscent of the diatomic Fe₂ system in catalyzing the epoxidation of trans-stilbene." The authors should discuss the role or absence of role of the two-coordinated O atom.*

Reply: We agree with and appreciate the Referee's valuable comments. In the revised version, we have carried out more calculations and added many new simulation results in the manuscript, including the entire process of styrene epoxidation on Pt₂ (Fig. 7). Here, since the reaction is carried out in an O₂ atmosphere, the active site is no longer the isolated dual Pt atoms, as in the cases of nitrobenzene hydrogenations. The active site now becomes an oxidized Pt₂O₂ species (S1), generated via the dissociation of an adsorbed O₂ molecule (S0 → S1 via TS1). The Pt₂O₂ species contains a one-coordinated oxygen atom and a two-coordinated oxygen atom, consistent with what we have observed in the EXAFS measurement. In the next few steps (from S2 to S4 in Fig. 7), the one-coordinated oxygen atom interacts with the C=C bond of the alkene and converts styrene to the corresponding epoxide. As the first styrene oxide molecule desorbs, there is only a two-coordinated oxygen atom locating at the Pt₂ site (S5). Such type of oxygen can be converted to a new one-coordinated oxygen atom by crossing an energy barrier of 0.62 eV (S5 → S6 via TS4), and then, participates in the next few steps of the epoxidation reaction (from S6 to S8). Upon the desorption of the second styrene oxide molecule as well as the adsorption of another O₂ molecule, the catalytic reaction cycle starts again. In Fig. 7, one can see that the one-coordinated oxygen atoms can directly participate in the epoxidation reaction, while the two-coordinated oxygen atom is not involved in the reaction until it is converted to the one-coordinated configuration. It is not surprising because the one-coordinated oxygen atom connects with Pt₂ via only one chemical bond and is thus less bound, which makes it easier to participate in the reaction and be grabbed by the styrene molecules. This phenomenon is very similar to the case of the diatomic Fe₂ system, which we used as dual-atom catalyst in the epoxidation of trans-stilbene molecules [S. Tian *et al.*, *Nat. Commun.* 9, 2353 (2018)].

The above discussion has been added in the revised manuscript.

Fig. 7. Reaction pathway and computational energy profile of styrene epoxidation on the $\text{Pt}_2/\text{g-C}_3\text{N}_4$ catalyst. The label S0 represents the initial state and the subsequent labels S1 – S9 represent a series of intermediate states. The labels TS1 – TS5 represent a series of transition states. Here, only key structures are shown and more details of the compositions and geometries are placed in the supplementary information (Supplementary Fig. S28). The teal, gray, blue, red, and white spheres represent the Pt, C, N, O, and H atoms, respectively.

Referee: In the section *Catalytic test* in page 11, the authors write: "The reactor was sealed then pressurized with 1 MPa H_2 and 3 MPa N_2 to a setting point." while in the main text in page 6, they write: "Under the conditions of 4 MPa H_2 pressure and 100□, a conversion of >99% was obtained on the $\text{Pt}_2/\text{mpg-C}_3\text{N}_4$ catalyst for the hydrogenation of nitrobenzene to aniline". (...) Please could you clarify which are the reaction conditions?

Reply: We thank the Referee for pointing this out. In the original manuscript, the expression "4 MPa H_2 pressure" is a mistake. We feel sorry for the mistake and have corrected it in the revised version on page 7: "Under the conditions of 1 MPa H_2 and 3 MPa N_2 pressure at 100 □, a conversion of >99% was obtained ...".

Referee: Following the same issue, in the following section: "Typical procedure for the hydrogenation of benzaldehyde", it is written: "The reactor was sealed, purged three times with N_2 at 1 MPa and then pressurized with 4 MPa H_2 and 4 MPa N_2 to a setting point." If the total pressure under reduction conditions is 8 MPa, they should write this in a more clear way, e.g. and then pressurized at 8 MPa of an H_2 and N_2 mixture (1:1) to a setting point.

Reply: We thank the Referee for the kind advice. In the revised version on Page 15,

we have rephrased the expression as follows: “The reactor was sealed, purged three times with 1 MPa of N₂ at and then pressurized at 8 MPa of an H₂ and N₂ mixture (1:1) to a setting point.”

Referee: Page 11: *"The products were identified by GC-MS and GC." The authors should give more details on the GC-MS and GC instruments used and also about the experimental conditions applied for these measurements.*

Reply: We thank the Referee for the kind advice. In the revised version on Page 16, we have added more information on the GC-MS and GC instruments: “The products were identified by the gas chromatography–mass spectrometry (GC-MS, Thermo Fisher Scientific-TXQ Quntum XLS), and were quantitatively analyzed by the gas chromatography (GC, Shimadzu, GC-2010 Plus), equipped with the flame ionization detector and a (30 m × 0.25 mm × 0.25 mm) KB-WAX capillary column (Kromat Corporation, USA) using n-octanol as the internal standard.”

The experimental conditions applied for these measurements are described on the same page as follows: “Operation parameters for the GC-MS measurements: The inlet temperature was 250 °C, the MS transfer line temperature was 250 °C, and the ion source temperature was 280 °C. The column temperature was first kept at 40 °C for 1 min, and then raised to 150 °C with a ramp rate of 20 °C/min, and was later raised to 250 °C with a ramp rate of 15 °C/min. Finally, it was kept at 250 °C for 7 min. Operation parameters for the GC measurements: The vaporization temperature and the detector temperature were both 270 °C. The column temperature was first kept at 40 °C for 1 min, and then raised to 240 °C with a ramp rate of 10 °C/min and was finally kept at 240 °C for 6 min.”

Referee: Page 20, Fig. 1d: *Could the authors explain features like the signaled in the attached picture? To get 100% Pt₂ structures or 100% Pt atoms supported sounds impossible and I miss more discussion about what the authors have found when studying the samples prepared by electron microscopy about the possibility to get also different structures in the different synthesis followed.*

Fig. 1. Characterization of Pt₂/mpg-C₃N₄ sample. (a) HAADF-STEM image of Pt₂/mpg-C₃N₄. (b) EDX mapping distributions of the C (blue), N (red), and Pt (green) elements, respectively. (c, d) AC HAADF-STEM images of the Pt₂/mpg-C₃N₄ and Pt₁/mpg-C₃N₄ samples, respectively.

Reply: We thank the Referee for the carefully reading and pointing out this issue. Very probably, the paired bright dot marked by the Referee in the original Fig. 1d come from two Pt single atoms, which were far away from each other in the three-dimensional space, but happened to be very close when being projected onto a certain two-dimensional plane, as shown in the Supplementary Scheme S1. This is because the AC HAADF-STEM image only represents a two-dimensional projection of a three-dimensional sample along the incident beam direction [Li, Z. Y. *et al. Nature* 451, 46-48 (2008)].

Scheme S1. The two-dimensional projection of three-dimensional sample along the incident beam direction in the AC HAADF-STEM characterization.

It should be noted that the emergence of such situation is very rare. Besides, since

there was no Pt–Pt path observed in the EXAFS measurements for the Pt₁/mpg-C₃N₄ sample, we could exclude the existence of the Pt₂ species in the the Pt₁/mpg-C₃N₄ sample. To avoid possible misunderstanding, in the revised version, we have replaced the original Fig. 1d with a new figure, where there is no paired bright dot. The original Fig. 1d has been placed in the Supplementary Fig. S10a, and in Fig. S10b, we have put a new AC HAADF-STEM image of the Pt₁/mpg-C₃N₄ sample obtained from another region.

Fig. S10. (a and b) AC HAADF-STEM images of the Pt₁/mpg-C₃N₄ sample obtained in different regions. There are paired bright dots in Fig. S10a as marked within the green rectangle.

Fig. 1d. AC HAADF-STEM images of the Pt₁/mpg-C₃N₄.

In the revised version, the above discussion has been added on pages 11-12 in supplementary information.

The Referee is absolutely correct that getting 100% Pt₂ or 100% Pt single atoms in the produced catalysts is very challenging. In this work, we argue that the obtained Pt species deposited on the mpg-C₃N₄ substrate is Pt₁ and Pt₂, respectively, based on the following reasons:

(1) In terms of the synthetic strategy, the precursor-preselected pyrolysis approach was employed to control over the formation of Pt₁/mpg-C₃N₄ and Pt₂/mpg-C₃N₄. First, H₂PtCl₆ and (Ethylenediamine)iodoplatinum(II) dimer dinitrate were selected as the mononuclear and binuclear precursors respectively, which is a prerequisite for the production of the Pt₁ and Pt₂ species. Then, the mesoporous graphitic carbon nitride (mpg-C₃N₄) was used as the substrate as it can provide abundant anchoring sites to

stabilize the metallic species. Finally, the pyrolysis process was carefully optimized to prevent the agglomeration of the Pt species.

(2) In terms of the structural characterizations, we fully agree with the Referee that to ensure exactly 100% Pt₂ or 100% Pt single atoms in the produced catalysts are almost impossible. In this work, we have tried our best to check and verify the statements about the structural patterns of the Pt species. Although technical “problems” such as incomplete imaging owing to an incomplete focusing could appear in the AC HAADF-STEM characterization, we have confirmed that the observed images in Fig. 1c indeed come from the dual-atom Pt₂ species, by comparing the different AC HAADF-STEM images that were collected within the same region of the sample but with the focus of the imaging constantly changed (Supplementary Fig. S6). In addition, since the special circumstances (shown in the Supplementary Scheme 1) is very rare, the existence of Pt₂ species in the Pt₁/mpg-C₃N₄ sample could mostly be excluded. Furthermore, we have also employed XAFS to support our statements. Concretely, the EXAFS data and the fitting results showed that the Pt₂/mpg-C₃N₄ sample exhibited Pt–Pt path with the coordination number being 1.1, whereas no Pt–Pt path was observed in the Pt₁/mpg-C₃N₄ sample.

Referee: Page 21, Fig. 2c. As I already said before, in the figure caption the authors mention: "The inset of c is the optimized geometry of Pt₂/mpg-C₃N₄." and there is no inset.

Reply: We thank the Referee for pointing this out. The corresponding figure is in Fig. 2f. We feel sorry for the mistake and have corrected it in the revised version.

Referee: Page 24, Fig. 4: The distances in b are difficult to see, please improve. Furthermore, I would suggest to use a different color for the Pt atoms.

Reply: We thank the Referee for the kind suggestion. After having gained a deep understanding of the reasons for the unique catalytic properties of the Pt₂/g-C₃N₄ system, we have completely rewritten the corresponding paragraphs and have deleted the original Fig. 4.

Referee: As a general comment, the characterization by transmission electron microscopy should be reinforced. The dual Pt structures are clearly shown, as expected, in the aberration corrected measurements but not in the other. Therefore, these measurements should also be carried out in the post-mortem catalysts. Furthermore, in situ XAS measurements would be appreciated and would give a plus to this work. Notwithstanding, the synthesis and the control of the dispersion of the catalytic entities is a big asset. As stated before, the characterization of the catalysts by AC-TEM measurements after the reaction in order to demonstrate that the dispersion is kept and there is no sintering is of key importance.

Reply: We agree with and appreciate the Referee's valuable comments. In the revised version, the AC STEM measurements have also been carried out for the post-mortem catalysts, with the corresponding results shown in the Supplementary Fig. S17 for Pt₂/mpg-C₃N₄ and in the Supplementary Fig. S19 for Pt₁/mpg-C₃N₄. It can be seen that the Pt species in Pt₁/mpg-C₃N₄ and Pt₂/mpg-C₃N₄ was still well dispersed as single atoms and dual-atoms, respectively, indicating that both Pt₁ and Pt₂ were stable in the catalytic process. As a note, the recycling time of the catalytic reaction has been extended from 5 cycles to 10 cycles in the revision.

Fig. S17. AC HAADF-STEM image of Pt₂/mpg-C₃N₄ after the reaction.

Fig. S19. AC HAADF-STEM image of the Pt₁/mpg-C₃N₄ after the reaction.

Furthermore, time-dependent XAFS measurements have also been performed in the revision to detect the changes in the oxidation state and chemical bonding of the Pt₂ species during the process of the nitrobenzene hydrogenation (Supplementary Fig. S21-22). The Pt *L*₃-edge in the XANES spectra of Pt₂/mpg-C₃N₄ showed that the intensity of the white line peaks became lower during the reaction (Supplementary Fig. S21), meaning that the oxidation state of Pt was smaller than that in the initial state. It is not surprising because the oxygen atoms attached to Pt can be removed by the hydrogen molecules. Besides, the EXAFS spectra showed that the first shell peak shifted from 1.57 Å to 1.55 Å (Supplementary Fig. S22), indicating that shorter chemical bonds like that of Pt-H appeared in the reaction process. This information has been added on Page 8 in the revised manuscript.

Fig. S21. Time-dependent XANES spectra of Pt₂/mpg-C₃N₄ during the hydrogenation of nitrobenzene to aniline.

Fig. S22. Time-dependent EXAFS spectra of Pt₂/mpg-C₃N₄ during the hydrogenation of nitrobenzene to aniline.

SUPPL. INFO:

Referee: page 1: Tian et al instead of tian ...

Reply: We thank the Referee for pointing this out and have corrected it in the revised version.

Referee: page 9: In the figure S8 caption, give a brief explanation of the inset as it was done in Fig. 2c caption (although in this case, the comment should be removed or the inset added since it is no displayed!)

Reply: We thank the Referee for the kind advice. The original Supplementary Fig. S8 has now been renumbered as Fig. S11 (on Page 13 in Supplementary information) in the revised version. In the corresponding figure caption, we have added the following explanation: “The inset is a schematic model of the Pt₁/g-C₃N₄ system. The teal, gray, blue, and red spheres represent the Pt, C, N, and O atoms, respectively.”

Referee: page 14, Fig. S12: Can you show the images before and after the reaction with the same scale? You should use the same scale also in Fig. S3 and it would be nice for the reader to see a direct comparison of both images, from before and after the reaction.

Reply: We thank the Referee for the very valuable suggestions. In the revised version, the original Fig. S12, which shows the TEM image of Pt NPs/mpg-C₃N₄ before the reaction, has been renumbered as the Supplementary Fig. S16. Besides, we have added the Supplementary Fig. S20 to display the corresponding TEM image of Pt NPs/mpg-C₃N₄ after the reaction. Here, we have used the same scale for the above two figures and also for the Supplementary Fig. S3.

Fig. S16. TEM image of Pt NPs/mpg-C₃N₄ before the reaction.

Fig. S20. TEM image of Pt NPs/mpg-C₃N₄ after the reaction.

Referee: page 15, Fig. S13, following my previous comment...you should show two pictures with the same scale, one before and one after the reaction to allow for a direct comparison.

Reply: We thank the Referee for the very valuable suggestions. In the revised version, the original Fig. S13, which shows the HAADF-STEM image of Pt₂/mpg-C₃N₄ after the reaction, has been replaced by a new figure showing the AC HAADF-STEM image of Pt₂/mpg-C₃N₄ after the reaction (renumbered as the Supplementary Fig. S17). We have applied the same scale in the Supplementary Fig. S17 (Pt₂/mpg-C₃N₄ after the reaction) as that in Fig. 1c (Pt₂/mpg-C₃N₄ before the reaction).

Besides, we have also added an AC HAADF-STEM image for the Pt₁/mpg-C₃N₄ sample after the reaction (Supplementary Fig. S19). The same scale has also been applied in this figure (Pt₁/mpg-C₃N₄ after the reaction) and in Fig. 1d (Pt₁/mpg-C₃N₄ before the reaction).

Fig. S17. AC HAADF-STEM image of Pt₂/mpg-C₃N₄ after the reaction.

Fig. S19. AC HAADF-STEM image of the Pt₁/mpg-C₃N₄ after the reaction.

Referee: page 17, Fig. S15, I would recommend to use a different color for the Pt atoms in order to get a better contrast.

Reply: We thank the Referee for the kind suggestion. After having gained a deep understanding of the reasons for the unique catalytic properties of the Pt₂/g-C₃N₄ system, we have completely rewritten the corresponding paragraphs and have deleted the original Figs. S15.

Referee: Apart from these comments/questions/suggestions, there are other typos to consider:

- page 6, 100□, 100 □ instead
- page 8: 120□, 120 □ instead
- page 25, figure 6 caption: hhydrogenation

Reply: We thank the Referee for pointing them out and have corrected the typos in the revised version.

Referee: *Although in the main text the authors have done a good job describing in a clear and concise way the different parts, in the section Materials and methods there are several typos and grammatical issues to fix: "In preparation of Pt nanoparticles, ", "for a further 1 hour", "showed almost no more weight was losing", "was dissolved in (the) 100 mL H2O", "in a 20 ml of Schleck tube", "was heated in (a) oil", etc.*

Reply: We thank the reviewer for pointing them out. In the revised version, we have carefully proofread the section Materials and methods and paid special attention to fixing the typos and grammatical issues. The corresponding changes are labelled in blue.

Reply to the Report of Reviewer 3

Referee: *The authors reported dual-atom Pt heterogeneous catalyst supported on mpg-C₃N₄ with excellent catalytic performances for the selective hydrogenation and epoxidation. The topic is very interesting. I am not an expert from experiment, but I can see the experimental synthesis and characterization have been carried out with full care. The dual-atom Pt/mpg-C₃N₄ catalyst has demonstrated the highest catalytic performance compared to single Pt/C₃N₄ atom catalyst. Overall the paper is well organized. I would like to recommend it publishing in Nature Communications after addressing the following minor issues.*

Reply: We thank the Referee for his/her carefully reading, very positive evaluation, insightful comments, and valuable suggestions. After having addressed the Referee's comments, the quality of the manuscript has been significantly improved.

Referee: *It would be good for authors to give more comparison between Pt₁, Pt₂ and Pt nanoparticles and provide a more clear picture on why the Pt₂/mpg-C₃N₄ catalyst can achieve high catalysis performance.*

Reply: We agree with and appreciate the insightful comments, which prompt us to explore the underlying reason for the unique and excellent catalytic properties of the dual-atom Pt₂ system. According to our analysis, the reason why Pt₂ is more efficient than the single atom or Pt nanoparticles in the nitrobenzene hydrogenation is that, not only the N-O bonds of nitrobenzene can be easily broken (assisted by the two Pt atoms) and hydrogenated on Pt₂, the desorption of the generated aniline product is also a facile process (promoted by the configuration change of a hydrogen adsorbate and the competitive adsorption of the nitrobenzene reactant), which helps the restart of the catalytic cycle. By contrast, however, the above two features are not simultaneously possessed in either Pt₁ or the Pt nanoparticles.

We first compare the difference between Pt₁ and the dual-atom catalyst. The reaction pathways and the computational energy profiles of nitrobenzene hydrogenation on Pt₂/g-C₃N₄ and Pt₁/g-C₃N₄ are displayed in Fig. 4 and Fig. 5, respectively. Here, since the active site of the Pt₁ system contains only one Pt atom, the occurrence of the N-O bond cleavage is not as easy as that on Pt₂, because at least two adsorption sites are required to stabilize the produced groups after the bond rupture (S1 → S2 via TS1 and S7 → S8 via TS4 in Fig. 4). Although the breaking of the N-O bond can also be assisted by the Pt atom and an adjacent C atom on the Pt₁ system, as shown in the S2 → S3 process via TS2 in Fig. 5, the presence of such C atoms cannot *always* be maintained during the reaction process. In the breaking of the second N-O bond S6 → S7 via TS4 (Fig. 5), for example, there is no neighboring C atom involved in and thereby, the corresponding energy barrier becomes as high as

2.31 eV. Thus, it is not surprising that the Pt₁ system cannot exhibit the same excellent catalytic properties as Pt₂.

Fig. 4. Reaction pathway and computational energy profile of nitrobenzene hydrogenation on the Pt₂/g-C₃N₄ catalyst. The label S0 represents the initial state and the subsequent labels S1 – S13 represent a series of intermediate states. The labels TS1 – TS6 represent a series of transition states. Here, only key structures are shown and more details of the compositions and geometries are placed in the supplementary information (Supplementary Fig. S23). The teal, gray, blue, red, and white spheres represent the Pt, C, N, O, and H atoms, respectively.

Fig. 5. Reaction pathway and computational energy profile of nitrobenzene hydrogenation on the Pt₁/g-C₃N₄ system. The label S0 represents the initial state and the subsequent labels S1 – S10 represent a series of intermediate states. The labels TS1 – TS5 represent a series of transition states. Here, only key structures are shown and more details of the compositions and geometries are placed in the supplementary information (Supplementary Fig. S24). The teal, gray, blue, red, and white spheres represent the Pt, C, N, O, and H atoms, respectively.

On the outermost layer of the Pt nanoparticles (simulated by a Pt(111) surface in the calculations), the hydrogenation is induced and assisted by the adsorbed H atoms that are produced via a spillover process. The overall energy barrier was calculated to be only 0.75 eV [T. Sheng *et al.*, *Chem. Eng. J.* 293, 337-344 (2016)], indicating that the occurrence of the hydrogenation is not difficult. The obstacle, however, comes from the desorption step of the aniline product. Our calculations show that the adsorption energy of aniline on Pt(111) is as high as -1.70 eV, and in the adsorption configuration, several C atoms of the phenyl group are involved in the bonding with the surface Pt atoms (Supplementary Fig. S25A). It means that upon aniline desorption, the product molecule needs to overcome a high energy barrier. In addition, since the adsorption energy of nitrobenzene (-1.35 eV) is smaller than that of aniline on Pt(111), the desorption of the aniline product cannot be promoted via the competitive adsorption of the reactant. Such the problem, however, does not appear in the Pt₂ system. Despite that the adsorption energy of aniline on Pt₂ is as high as -2.58 eV, the adsorption of the reactant is stronger, showing an adsorption energy of -3.16 eV. Moreover, in this case, there is only one N-Pt bond connecting aniline with the Pt₂ catalyst (Supplementary Fig. S25C), and the aniline desorption can also be facilitated by a change in the configurations of the hydrogen adsorbate on Pt₂ (as shown in the S10 → S11 process in Fig. 4). Thus, the aniline desorption can be easily achieved and is no longer an obstacle on Pt₂.

Fig. S25. Adsorption configurations of an aniline molecule (A and C) and a nitrobenzene molecule (B and D) on the Pt(111) surface (top) and the Pt₂/g-C₃N₄ catalyst (bottom). The corresponding adsorption energy values are displayed in the respective lower right corners. The teal, gray, blue, red, and white spheres represent the Pt, C, N, O, and H atoms, respectively.

We have added the discussion (on pages 8-11 in the main text), Figs. 4-5 (in the main text) and Fig. S25 (in the supplementary information) in the revised vision.

Referee: Can authors provide more recycling times for the $Pt_2/mpg-C_3N_4$ catalyst?

Reply: We thank the Referee for the kind advice. To further demonstrate the stability of the $Pt_2/mpg-C_3N_4$ catalyst in the nitrobenzene hydrogenation, we have extended the recycling time from the original five cycles to ten cycles. It was found that the $Pt_2/mpg-C_3N_4$ sample did not lose any activity during the time. Moreover, for the hydrogenation of benzaldehyde and epoxidation of styrene, the recycling time were set to five cycles, and the catalyst did not lost its activity either. The corresponding results are placed in Fig. 3b and Fig. 6 in the main text.

Fig. 3. (b) Recycling of $Pt_2/mpg-C_3N_4$ for the catalytic hydrogenation of nitrobenzene.

Fig. 6. Hydrogenation of benzaldehyde and epoxidation of styrene. (a) Catalytic performance for the hydrogenation of benzaldehyde by using the $Pt_2/mpg-C_3N_4$ catalyst. (b) Corresponding catalytic performance for the epoxidation of styrene.

Referee: The distortion of $Pt_2/mpg-C_3N_4$ is highly expected. Can authors give some comments on it?

Reply: The Referee is absolutely correct that the $Pt_2/mpg-C_3N_4$ plane is distorted with

obvious undulations. In the revised version, we have explicitly mentioned it in the main text on Page 6 and have placed a side view of the Pt₂/g-C₃N₄ structure in the Supplementary Fig. S13 in the supplementary information.

Fig. S13. Side view of Pt₂/g-C₃N₄ structure, showing the distortion of the g-C₃N₄ substrate with obvious undulations. The teal, gray, and blue spheres represent the Pt, C, and N atoms, respectively.

Referee: *Some recent theoretical works on the development of Pt/g-C₃N₄ based catalysts for the hydrogenation and reduction reactions could be cited [e.g. JACS 138 (2016) 6292; Nano Research 12 (2019) 1817].*

Reply: We thank the Referee for pointing out the references. We have carefully read the papers and have cited them properly in the revised version as follows:

24. Gao, G. Jiao, Y. Waclawik, E. & Du, A. Single Atom (Pd/Pt) Supported on Graphitic Carbon Nitride as an Efficient Photocatalyst for Visible-Light Reduction of Carbon Dioxide. *J. Am. Chem. Soc.* **138**, 6292-6297 (2016).

25. He, T. Zhang, C. Zhang, L. & Du, A. Single Pt atom decorated graphitic carbon nitride as an efficient photocatalyst for the hydrogenation of nitrobenzene into aniline. *Nano Res.* **12**, 1817-1823 (2019).

Reviewers' comments:

Reviewer #1 (Remarks to the Author):

The authors of the manuscript "Dual-atom Pt heterogenous catalyst with excellent catalytic performances for the selective hydrogenation and epoxidation" have addressed all the points raised by the present reviewer especially concerning the description of the mechanism for the hydrogenation of the nitrobenzene. This is the second time that I review this manuscript. Although the authors have carried out a very detailed computational investigation, I want to recall that the computational analysis should not be a simple appendix or a collection of numbers but it must provide some useful hints for better understanding the mechanism of the reaction and the correlations between electronic and structural features associated to each step of the process. In this case, the presentation of the computational results is very confused since the authors reported a tedious sequence of Intermediates and Transition State (in many cases without any chemical meaning) without any correlations with the electronic structure. In addition, the reported Figures do not help in understanding the process in details. I want to recall that for each transition state the authors should provide the associated vibrations together with all the structural features. The authors wrote that the geometries and structural features are reported in the Supporting Information but in the SI they report only the same figures of the manuscript without any description or lists of coordinates. This is not acceptable for a serious work.

Focusing on Figure 4, what about the H-H distance at S3 and TS2 compared to the free H₂? How elongated is the H-H distance? And the associated vibration? It seems quite strange that heterolytic splitting of the H₂ could occur between an oxygen and a nitrogen centres without the coordination of the hydrogen molecule to the metal. Did the authors take into account this? Another point concerns the quite easy breaking of the N-O bonds along the pathway without any reasonable explanation of the nature of the products or correlation with the electronic structure. Is it a homolytic or heterolytic cleavage?

The situation seems also worst for the single atom catalyst, in Figure 5, for which the authors predicted the formation of OH moiety, then bound to a carbon centre of the support and, after the improbable interaction with the incoming H₂, a water molecule is generated. Is the generated OH a radical species or a OH⁻ anion? What about the electronic consequences on the Carbon atom of the support bound to the OH and on the overall support?

IN addition, I have many doubts on the plausibility of the correlations of only 0.02 Å variation in the first shell peak in EXAFS with the reactivity as reported at page 8 line 170.

For all these reasons, the manuscript is in no way suitable for a top journal like Nature Communications and, before the submission to another journal, the authors must strongly review and think about other possible and, in some cases, more chemical plausible, mechanisms.

Reviewer #2 (Remarks to the Author):

The authors have provided answers and solutions for all the points raised during the previous review of this work. The quality of this article has been improved by assessing all the questions from the reviewers and now is more clear and consistent, therefore, I recommend its publication in Nature Comm.

Reviewer #3 (Remarks to the Author):

The authors basically have carefully addressed all the concerns raised by the reviewers. Now it can be acceptable for publication in Nature Communications.

Response to the Referee's Comments

Reply to the Report of Reviewer 1

Referee: The authors of the manuscript “Dual-atom Pt heterogeneous catalyst with excellent catalytic performances for the selective hydrogenation and epoxidation” have addressed all the points raised by the present reviewer especially concerning the description of the mechanism for the hydrogenation of the nitrobenzene. This is the second time that I review this manuscript. Although the authors have carried out a very detailed computational investigation, I want to recall that the computational analysis should not be a simple appendix or a collection of numbers but it must provide some useful hints for better understanding the mechanism of the reaction and the correlations between electronic and structural features associated to each step of the process. In this case, the presentation of the computational results is very confused since the authors reported a tedious sequence of Intermediates and Transition State (in many cases without any chemical meaning) without any correlations with the electronic structure.

Reply: We are pleased to see that the Referee has recognized the efforts that we had made in the previous round of revision, and we are very grateful that in the second round of reviewing, the Referee has explicitly pointed out detailed and very insightful comments regarding how to improve the computational simulations. In this round of revision, we have followed the suggestion of the Referee to establish correlations between the electronic structures and the associated reaction steps, such as in the activation of the N-O bonds upon nitrobenzene adsorption and in the dissociation of the H₂ reactant around an O and an N atoms (both of them appear in Fig. 4). Through these analysis, we have found that the N-O bond activation originates from the partial occupation of the lowest unoccupied molecular orbital (LUMO) of nitrobenzene when it is adsorbed on Pt₂, and have also revealed the role of the unsaturated N and O adsorbates that are generated upon the N-O bond cleavage on Pt₂ in the H₂ dissociation. The activation of the N-O bonds and the subsequent generation of the unsaturated N and O adsorbates benefit from the diatomic characteristics of the Pt₂ species, which is an important reason why Pt₂/g-C₃N₄ exhibits unique and excellent catalytic performance in nitrobenzene hydrogenation. In the revised version, we have

expanded the discussion about the simulation results. The quality of this work has been further improved.

Referee: In addition, the reported Figures do not help in understanding the process in details. I want to recall that for each transition state the authors should provide the associated vibrations together with all the structural features.

Reply: We thank the Referee for this very helpful suggestion. For each transition state, we have listed the imaginary frequency value and displayed the associated animation for the corresponding vibrational mode, provided in the attached file “vib-TS.pptx” (Please visualize the file using the “Slide Show” mode) as the supplementary information. It may be worth noting that we have paid special attention in analyzing the corresponding vibrational mode, and have confirmed that each transition state is indeed connected to the correct initial and final states of the elementary reaction step. In the revised manuscript, we have added the following sentences: “Each transition state has been confirmed via the vibrational mode analysis to ensure that it is indeed connected to the correct initial and final states of the elementary reaction step. All the imaginary frequencies and the animations related to the associated vibrational modes are provided in the attached file “vib-TS.pptx” (visualized in the “Slide Show” mode) as the supplementary information.”

We would like to mention that by following the Referee’s suggestion to analyze the vibrations of the transition states, we have found an energetically more favorable way for the dissociation of the third H₂ molecule on Pt₁/g-C₃N₄ (S6 → S7 via TS4 in Fig. 5). From this result, we find that the dissociation of the H₂ reactant is more critical than the N-O bond cleavage in explaining the difference in the catalytic performance toward nitrobenzene hydrogenation between Pt₁/g-C₃N₄ and Pt₂/g-C₃N₄. In the new **Abstract**, the corresponding expression have been changed to “the excellent and unique catalytic performance of the Pt₂ species originates from the facile H₂ dissociation induced by the diatomic characteristics of Pt and the easy desorption of the aniline product”. Besides, in the revised manuscript, we have discussed more about the participation of C atoms from the g-C₃N₄ framework in the reaction on Pt₁/g-C₃N₄: “By comparing the reaction pathways in Figs. 4 and 5, one can see that for the Pt₁/g-C₃N₄ system, the g-C₃N₄ framework not only serves as a substrate to anchor the Pt atoms, but also, its C atoms can directly participate in the nitrobenzene hydrogenation. For example, in the cleavage of the first N-O bond on Pt₁/g-C₃N₄ (Fig.

5), an adjacent carbon atom to Pt acts as the adsorption site to stabilize the produced OH radical ($S2 \rightarrow S3$ via TS2). In the activation and dissociation of the third H_2 molecule ($S6 \rightarrow S7$ via TS4), this carbon atom, together with the oxygen atom nearby, stabilize the two produced hydrogen atoms.” We thank the referee again for the very helpful suggestion.

Referee: The authors wrote that the geometries and structural features are reported in the Supporting Information but in the SI they report only the same figures of the manuscript without any description or lists of coordinates. This is not acceptable for a serious work.

Reply: It seems that our arrangement of the structural information (in Figures of the main text and the supplementary information) had caused misunderstanding. In the main text, only the key structures in each elementary step, *i.e.*, the Pt_2/Pt_1 catalytic system as well as the adsorbate bound on it, are displayed in the Figures. All other species, like the reactant molecules which have not been adsorbed and/or the product molecules which have been desorbed, are not shown due to a limitation of space. Thus, all the missing information has been added in the corresponding supplementary Figures in the SI. Here, we emphasized that it is necessary and important to supplement such information, because when one directly compares the energies of different elementary steps, each step must contain exactly the same type and number of atoms. Thus, the supplementary Figures in the SI are not simply repetitions of the Figures in the main text.

To clarify this point, in the revised version, we have rephrased the corresponding figure captions (for the Figs. 4, 5, and 7, and the supplementary Fig. S29) as “Here, only the key structures, *i.e.*, the $Pt_2(Pt_1)$ catalytic system as well as the adsorbate bound on it, are shown. The information regarding reactant molecules which have not been adsorbed and/or product molecules which have been desorbed are labelled in the supplementary information.”

Besides, following the Referee’s suggestion, information of the coordinates (*i.e.*, the POSCAR files) for all the structures, excluding those of simple small molecules like H_2 , have been provided at the last section of the revised supplementary information. In the revised manuscript, we have added the following sentences: “The POSCAR files for all the optimized structures are placed in the last section of the supplementary information.”

Referee: Focusing on Figure 4, what about the H-H distance at S3 and TS2 compared to the free H₂? How elongated is the H-H distance? And the associated vibration? It seems quite strange that heterolytic splitting of the H₂ could occur between an oxygen and a nitrogen centres without the coordination of the hydrogen molecule to the metal. Did the authors take into account this?

Reply: According to our simulations, the H-H distance at S3 and TS2 are 0.748 Å (the elongation of 0.004 Å is negligible) and 0.819 Å (elongated by 0.075 Å), respectively, compared with that of 0.744 Å of an isolated H₂ molecule. The corresponding imaginary frequency is calculated to be 596.4 cm⁻¹. Again, we have confirmed that the TS2 is indeed connected to S3 and S4 on the potential energy surface, since the vibrational mode corresponds to an elongation of the H-H bond (although the bond elongation is not so obvious in the animation due to the small increase of the bond length in TS2) and an approaching of these two H atoms towards the O and the N centers, respectively.

We fully understand the Referee's concern about the correctness of the transition state (TS2) and the corresponding reaction mechanism. Honestly, the reaction pathway of coordinating H₂ to the Pt site, as mentioned by the Referee, had indeed been our initial consideration on the reaction profiles. Although such a way of activating the hydrogen molecule obeys the traditional concept, it was excluded by our calculations, since the energy barrier of an H atom diffusing to the O/N atom is much higher than the value based on TS2. Thus, the reaction pattern without the involvement of metal, as found here, may represent a new mechanism. It is reminiscent of the nitrogen-doped carbon nanotube arrays with high electrocatalytic activity for oxygen reduction [Dai et al., Science 323, 760 (2009)], which had opened a new research area of metal-free electrocatalysis. In the revised version, we have added the following sentence: “Here, the activation of the H₂ molecule does not involve the participation of Pt atoms, reminiscent of the nitrogen-doped carbon nanotube arrays as metal-free electrocatalysts for the oxygen reduction reaction”.

To shed light on the mechanism of H₂ activation in this elementary reaction step, we analyze the electronic structures of the S3 and the TS2 configurations (supplementary **Figure S24**). While in S3 (the left panel), the two H atoms have the same electronic structure that is very close to that of the hydrogen atoms in an isolated H₂ molecule, in TS2 (the right panel), the electronic structures of the two H atoms are

significantly different. In particular, at the peaks of 0.39 eV and 0.09 eV below the Fermi level, the H atom close to the N atom (with the H-N distance being 1.87 Å) exhibits an obvious electronic state distribution, while the H atom close to O (with the H-O distance being 1.52 Å) has almost no distribution. Such remarkable contrast can also be visualized from the spatial distributions of the electronic states within the corresponding energy intervals (inset in the right panel). Here, one of the two H atoms as well as the unsaturated N and O adsorbates exhibits the distributions. The results indicate that the H₂ activation is facilitated by the produced N and O atoms and is promoted by a polarization effect induced by O. The latter is further supported by the Bader charge analysis, showing that in TS2, the two H atoms carry charges of +0.20 (close to O) and -0.08 (close to N), respectively. The above discussion has been added in the revised manuscript.

Figure S24: Calculated electronic density of state (DOS) of the two H atoms (H1 and H2) in the S3 (left panel) and TS2 (right panel) configurations in Fig. 4, corresponding to the initial state and the transition state of the H₂ dissociation step. The teal, gray, blue, red, and white spheres represent the Pt, C, N, O, and H atoms, respectively.

Referee: Another point concerns the quite easy breaking of the N-O bonds along the pathway without any reasonable explanation of the nature of the products or correlation with the electronic structure. Is it a homolytic or heterolytic cleavage?

Reply: We thank the Referee for the insightful suggestion, *i.e.* to gain an understanding of the N-O bond breaking mechanism through an analysis of the electronic structure. In the supplementary **Figure S25**, we present the calculated electronic density of state (DOS) of Pt₂/g-C₃N₄ upon the adsorption of a nitrobenzene molecule. One can see that the Pt diatomic species and the nitrobenzene adsorbate interact with each other at the Fermi level, and by analyzing the corresponding spatial distribution, we find that the electronic state comes from the lowest unoccupied

molecular orbital (LUMO) of nitrobenzene. Since the LUMO of nitrobenzene involves the anti-bonding interactions of the π orbitals between the N and the two O atoms (the inset in supplementary **Figure S25**), the fact that this LUMO appears at the Fermi level and is partially occupied will inevitably lead to the weakening of the N-O bonds, thereby resulting in the easy breaking of these bonds. The above discussion has been added in the revised manuscript.

Figure S25: Calculated electronic density of state (DOS) of $\text{Pt}_2/\text{g-C}_3\text{N}_4$ upon the adsorption of nitrobenzene. The LUMO orbital of an isolated nitrobenzene molecule is placed in the inset.

Our Bader charge analysis show that the N and the O atoms carry charges of +0.15 and -0.42, respectively, upon the adsorption of nitrobenzene (at the initial state S1). At the transition state (TS1), the two charge values change to -0.17 and -0.61, and finally become -0.14 and -0.50 after the N-O bond cleavage (at the final state S2). Thus, the rupture of the N-O bond is more like a heterolytic cleavage, accompanied (maybe also assisted) by charge transfer from the $\text{Pt}_2/\text{g-C}_3\text{N}_4$ catalyst.

Referee: *The situation seems also worst for the single atom catalyst, in Figure 5, for which the authors predicted the formation of OH moiety, then bound to a carbon centre of the support and, after the improbable interaction with the incoming H_2 , a water molecule is generated. Is the generated OH a radical species or a OH^- anion? What about the electronic consequences on the Carbon atom of the support bound to the OH and on the overall support?*

Reply: We think that the generated OH bound to the carbon atom is a radical rather than an OH^- anion. According to the optimized geometry (in Fig. 5), the N-O (in S2) and C-O (in S3) bond lengths are 1.50 Å and 1.44 Å, respectively, which are within

the scope of the covalent bond between the N/C and the O atoms. If the OH moiety were an anion, the OH moiety should be well saturated, and thus, the distance between N/C and the O atom would be much larger than the calculated value of 1.5 Å, since in this case, the OH group will bind to the g-C₃N₄ substrate via the electrostatic interactions.

Figure S28: Calculated electronic density of state (DOS) of the carbon atom bound to OH (left panel) and the entire g-C₃N₄ framework (right panel) for the configurations S2 (before OH connects to the C atom), S3 (with OH bound to the C atom), and S5 (after OH leaves the C atom) in Fig. 5.

In the supplementary **Figure S28**, we present the calculated electronic density of state (DOS) of the mentioned C atom (left panel) and the entire g-C₃N₄ framework (right panel) for the configurations S2 (before OH connects to the C atom), S3 (with OH bound to the C atom), and S5 (after OH leaves the C atom). One can see that when the OH group does not form a bond with the C atom, no matter the system adopts the configuration S2 or S5, the electronic structures of both the C atom and the g-C₃N₄ framework are not much different. However, when the OH radical is attached to the C atom, the electronic structures undergo obvious changes: For the C atom, the unoccupied states within 3 eV above the Fermi level disappear; while for the g-C₃N₄ framework, the entire DOS undergoes a right shift relative to the Fermi level. The changes in the above electronic structures also shows that the OH group has a strong interaction with the C atom, further supporting that OH is a radical rather than an anion. The above discussion has been added in the revised manuscript.

Referee: *IN addition, I have many doubts on the plausibility of the correlations of only 0.02 Å variation in the first shell peak in EXAFS with the reactivity as reported at page 8 line 170.*

Reply: We thank the Referee for the comment. In the manuscript, we describe that “The EXAFS spectra showed that the first shell peak shifted from 1.57 Å to 1.55 Å (Supplementary Fig. S22), indicating that shorter chemical bonds like that of Pt-H appeared in the reaction process.” Herein, it is worth mentioning that the EXAFS has a high sensitivity to the bond length change between center metal atoms and neighboring atoms, even when the change is as small as 0.01 Å (*Sun, Z., Liu, Q., Yao, T., Yan, W. & Wei, S. X-ray absorption fine structure spectroscopy in nanomaterials. Sci. China Mater. 58, 313–341 (2015)*). Although the shift of the first shell peak during the nitrobenzene hydrogenation is slight, it can indeed be monitored by EXAFS. The discussion has been added in the revised manuscript.

Referee: For all these reasons, the manuscript is in no way suitable for a top journal like Nature Communications and, before the submission to another journal, the authors must strongly review and think about other possible and, in some cases, more chemical plausible, mechanisms.

Reply: We thank the Referee for taking the time to read and review our manuscript. We think that the comments by the Referee in the first round are indeed insightful and very constructive. The new suggestions in the second round, like performing analysis of the electronic structures and the vibrational modes, are also very helpful. Following the Referee’s suggestions, we have added more calculations, verifications, and analysis in this round of revision, and have worked to clarify possible misunderstandings in the previous expressions. The quality of this work has been further improved.

Besides, we would like to mention that all the reaction pathways reported in the manuscript (in Figs. 4, 5, and 7, and in the Supplementary Fig. S29) were carefully determined after having made many attempts. Each transition state has been further confirmed via the vibrational mode analysis to ensure that it is indeed connected to the correct initial and final states of the elementary reaction step. We hope that our response to the comments could help the Referee understand and evaluate this work, and we sincerely hope that the Referee could now find the newly revised manuscript suitable for publication in *Nature Communications*.

REVIEWERS' COMMENTS

Reviewer #1 (Remarks to the Author):

The manuscript "Dual-atom Pt heterogeneous catalyst with excellent catalytic performances for the selective hydrogenation and epoxidation" by Wang et al. has been already considered by the present reviewer as not suitable for publication on Nature Communications. Although the authors have modified the manuscript in agreement with my precedent suggestions, the authors do not provide any strong evidence of the proposed mechanism of the catalytic hydrogenation, which, as already pointed out by the present reviewer, seems particularly strange. In addition, the paragraphs introduced in the text for answering the requests of the reviewer are very confusing and need more clarification. For all the aforementioned observations, the manuscript is not suitable for publication on Nature Communications and I strongly suggest again to the authors to think again about the mechanism before submission. The computational part is not a simple appendix of the experimental one but must provide explanations and confirmation to the proposed mechanism. All of this is not provided in the present manuscript.

Response to the Referee's Comments

Reply to the Report of Reviewer 1

Referee: The manuscript “Dual-atom Pt heterogeneous catalyst with excellent catalytic performances for the selective hydrogenation and epoxidation” by Wang et al. has been already considered by the present reviewer as not suitable for publication on Nature Communications. Although the authors have modified the manuscript in agreement with my precedent suggestions, the authors do not provide any strong evidence of the proposed mechanism of the catalytic hydrogenation, which, as already pointed out by the present reviewer, seems particularly strange. In addition, the paragraphs introduced in the text for answering the requests of the reviewer are very confusing and need more clarification. For all the aforementioned observations, the manuscript is not suitable for publication on Nature Communications and I strongly suggest again to the authors to think again about the mechanism before submission. The computational part is not a simple appendix of the experimental one but must provide explanations and confirmation to the proposed mechanism. All of this is not provided in the present manuscript.

Reply: We thank the Referee again for taking the time to read and review our manuscript. We are happy to see that our modification of the manuscript is considered by the Referee to be in agreement with his/her precedent suggestions. In the past two rounds of the revision, we have fully addressed all the comments of the Referee. The mechanisms proposed by the theoretical simulations have revealed key and in-depth information and provided good explanations for the experimental findings. We also realize that the Referee has even higher demands on our work, which are not possible to fulfill by anybody in the fields. The transition states determined by the calculations have already been carefully checked from a computational point of view, which, unfortunately, cannot be fully verified by experiments due to the lack of proper experimental techniques. It is a challenge not only for us, but also for all scientific community. We believe that we have done all we can for this work.